# Performance Analysis and Improvement for CRUD Operations in Relational Databases from Java Programs Using JPA, Hibernate, Spring Data JPA

Alexandru Marius Bonteanu [1] and Cătălin Tudose [1,2,*]

1   Faculty of Automatic Control and Computers, National University of Science and Technology Politehnica Bucharest, 060042 Bucharest, Romania; alexbonteanu22@gmail.com
2   Luxoft Romania, 060042 Bucharest, Romania
*   Correspondence: catalin.tudose@gmail.com

**Abstract:** The role of databases is to allow for the persistence of data, no matter if they are of the SQL or NoSQL type. In SQL databases, data are structured in a set of tables in the relational database model, grouped in rows and columns. CRUD operations (create, read, update, and delete) are used to manage the information contained in relational databases. Several dialects of the SQL language exist, as well as frameworks for mapping Java classes (models) to a relational database. The question is what we should choose for our Java application, and why? A comparison of the most frequently used relational database management systems, mixed with the most frequently used frameworks should give us some guidance about when to use what. The evaluation is conducted based on the time taken for each CRUD operation to run, from thousands to hundreds of thousands of entries, using the possible combinations in the relational database system and the framework. Aiming to assess and improve the performance, the experiments included the possibility of warming-up the Java Virtual Machine before the execution of queries. Also, the research investigated the time spent using different methods of code to determine the critical regions (bottlenecks). Thus, the conclusions provide a comprehensive overview of the performances of Java applications accessing databases depending on the suite decisions considering the database type, the framework in use, and the type of operation, with clear comparisons between the alternatives, the key findings of the advantages and drawbacks of each of them, and supporting architects and developers in their technological decisions and improving the speed of their programs.

**Keywords:** Java; relational databases; CRUD operations; Java Persistence API; Hibernate; Spring Data JPA; database performance

## 1. Introduction

Accessing relational databases from Java applications is the function of the application domain. It is essential for any application to persist its data. One possibility for doing this is by directly creating SQL queries [1] in the Java code [2,3] and manually mapping the classes and their attributes to the tables and their columns. This work is tedious, and there is also a lack of portability: when switching to another database with a different SQL dialect, the queries need to be rewritten. A different approach uses Object Relational Mapping (ORM), which concerns itself with the mapping of classes to database tables and forms a layer that hides this work from the developer, allowing him to focus on the programming logic [4].

There are a lot of frameworks and SQL dialects that can be used when you start a new application, but this makes the choice harder. The complexity and diversity of the alternatives require some initial evaluation that may be based on the amount of code to be written, the dependencies needed in the project, and the execution speed [5]. Each of them has its own advantages and disadvantages, in the end coming to a trade-off, as frequently happens in information technology. Using one method, we may have to consider speed in

accessing the database; using another method, we may have to consider portability, the amount of code to be written, and the development speed. A developer should be aware of his application's needs to be able to pick the right combination, generally being able to choose, as a framework, between JPA, Hibernate [6,7], and Spring Data JPA [8].

This paper expands upon previously published work and experiments, summarizes them, and focuses on improving the execution speed and detecting bottlenecks. This will help developers to choose the option that best fits their Java [2,3] application. Picking a framework randomly might not be the best solution to start an application.

The proposed research approach is to run an average-complexity application with different frameworks and dialects on a Windows operating system and to test the necessary time to handle different numbers (up to 500 thousand entries) of each CRUD operation. The amount of code and its simplicity will also be taken into consideration when the comparison is made.

The research approach includes the following steps:

1.  The design of the database that will serve as the experimental support.
2.  The design and implementation of the Java program that will interact with the database. This includes the Java classes and tests that use different frameworks to work with the database (JPA, Hibernate, Spring Data JPA).
3.  Running a Java test will be completed on a single thread, using combinations of all previously mentioned frameworks and different relational database servers (MySQL using the InnoDB engine, Oracle, PostgreSQL, and Microsoft SQL Server), and we will present the experimental results in tables and graphics.
4.  Improving performances using JMH with one warm-up iteration and one execution iteration, running Java tests using combinations of all the mentioned frameworks and database servers, collecting the experimental results in tables and graphics, and comparing them with previous results.
5.  Identifying the bottlenecks and the code sections that require the most time to be executed.

MySQL [9] is an open-source RDBMS with a large user base. It is widely used for developing web applications, as it has a low cost and simple installation process. Its capabilities include replication, full-text search, and support for different storage engines. InnoDB [10–12] is the default MySQL storage engine.

Oracle [13] is largely used for big businesses. The reasons for this are its robustness and scalability. Oracle provides a range of data warehousing and analytics tools but has high licensing costs and hardware requirements.

PostgreSQL [14,15] is also an open-source RDBMS. It has extensible functionalities, providing its own data types and functions. It is a frequent choice for data science, offering support for machine learning and analytics.

SQL Server [16,17] is a proprietary Microsoft product, largely used in applications that interact with other Microsoft products, with which it integrates very well. It includes features for data warehousing and analytics, such as columnstore indexes, and in-memory OLTP. SQL Server provides a free version for small databases and a paid version for large businesses.

## 2. Related Work

Several studies investigate the performances of individual databases, or present which SQL dialects or ORM frameworks the developer should choose and why. The investigation evolved from the analysis of individual databases and the necessary steps that need to be taken to ensure that a DBMS performs optimally [18] in comparison to relational and non-relational databases, considering the need to choose one of the alternatives for particular conditions. The comparison criteria include theoretical differences, features, restrictions, integrity, distribution, system requirements, architecture, queries, and insertion times [19].

Particular databases provide improvement possibilities, for example Oracle's optimizer hints that it may force various approaches for reducing the execution times of stored procedures. Some system parameters have greatly influenced the performance of the execution times of stored procedures and functions [20].

Another strategy is optimizing the execution plan, analyzed in depth in [21], using an application of mobile telephony through the execution of the stored procedure PL/SQL, which processes telephone invoices on a database with a million recordings.

The existence of various performance optimization alternatives generated the idea of classifying them as prediction, diagnosis, and tuning approaches. The prediction of future performance can influence changes in configurations and resources. The analysis of anomalies can diagnose the ground cause introducing performance regression. Tuning operations improve the performance by adjusting influencing factors such as the indexes, views, stored procedures, and queries' designs [22].

Current trends in Big Data have generated more interest in the in-depth analysis of tuning possibilities. Obtaining quick results is limited by the speed of big networks and their data processing capabilities. Finding the data location and the dynamic modification of the current data generates problems in fetching the repository. Consequently, scaling the databases requires tuning the distributed framework and programming languages to process large datasets over the network [23].

The vast landscape of the database world and the advantages of NoSQL has generated interest in analyzing in depth their performances. The widely used open-source NoSQL document databases, Couchbase, CouchDB, and MongoDB are evaluated in [24] using the Yahoo! Cloud Serving Benchmark (YCSB) [25], which has become a standard for NoSQL database evaluation.

Data representation in different formats and the data exchange necessity over the internet require analyses of the performances of the efficiency of mapping techniques, which need to be executed in two ways, before and after the transmission of data [26].

The rise of Cloud Computing creates particular performance challenges. The Cloud Provider intends to offer database services that are able to control the response time for specific customers. Popular databases that are widely used may be adapted to enable differentiated per-user/request performance on a priority basis through CPU scheduling and the synchronization mechanisms available within the operating system [27].

Database access optimization may also rely on the initial performance evaluation and on improving the performances through machine learning. The strategy is to first calculate consumption, then optimize the main performance modules and select the optimized structure of the database for calculation, and conduct a data analysis for the calculation results [28].

Haseeb Yousaf compares, while experimenting with the Ubuntu operating system, several ORM frameworks: Hibernate, OpenJPA, and EclipseLink. He performed five queries to access a table. The first query made a read using the primary key; three queries made a read using different type attributes; and one query made a read using two attributes. The experiments demonstrated that, for a large range of record numbers (10,000 to 160,000), Hibernate is the fastest and OpenJPA is the slowest. While increasing the number of records, EclipseLink moves closer to the speed of Hibernate, but this maintains the best performance [29].

Hossain, Sazal, and Das Santa compare MySQL, Oracle, PostgreSQL, SQLite, and Microsoft SQL Server [30]. Oracle proves to have low performance for reading from the database, but it is better for updating and deleting. On the contrary, MySQL has low updating and deleting performances, while PostgreSQL is the fastest one for these operations. SQLite and Microsoft SQL Server may serve as alternatives to PostgreSQL.

The comparisons mentioned above are made at the same level of technology, either to test different ORM frameworks or to measure the performances of RDBMSs.

The research we propose covers the suite of all the technologies needed for an application and achieves a comprehensive evaluation, including combinations of the ORM framework and the SQL database.

A research study using the .NET programming language was conducted at the University of Oradea [31], where the analysis compared both execution times and memory usage of an application with SQL Server as the RDBMS and switching between three distinct ORM frameworks.

Another research study for .NET was conducted at the Faculty of Electrical Engineering from Niš by Stevica Cvetković and Dragan Janković [32]. This study included the reduced translation overhead that modern ORM tools had in the .NET environment.

Recent studies of the performances of ORM usage in Java were conducted by Colley, Stanier, and Asaduzzaman to examine the impact of this additional layer compared to direct access to databases using JDBC [33]. Individual studies of the Hibernate framework were conducted by Babu and Gunasingh [34] and by Alvarez-Eraso and Arango-Isaza [35]. Vaja and Rahevar studied the topic of performance through in-memory caching [36].

Our research differentiates itself from the previously published studies by covering all combinations of several ORM frameworks (JPA, Hibernate, and Spring Data JPA) and several databases (MySQL, Oracle, PostgreSQL, Microsoft SQL Server). Alongside this, our research investigates the critical execution sections (bottlenecks) and the improvements that the JVM warm-up brings.

### 3. Problem Background

Relational databases organize data using tables. Each record in the table must have a primary key (PK) that uniquely identifies it. A table contains data attributes called columns. Foreign keys (FK) are used to connect the related tables [37].

There are four principles to define relational database transactions:

- Atomicity—ensures that the transaction operations are all executed or none are;
- Consistency—ensures that only correct data are added to the database;
- Isolation—ensures that transactions are not affected by other transactions;
- Durability—ensures that data committed to the database is stored permanently.

Atomicity is essential as several statements are frequently included in transactions. Each transaction is handled as a separate "unit" that must either succeed fully or fail completely for atomicity to be maintained. Should any of the statements that make up a transaction fail, the transaction as a whole fails and the database remains untouched. An atomic system needs to ensure atomicity under all circumstances, such as faults, crashes, and power outages. A guarantee of atomicity prevents incomplete database updates from happening, which can lead to more issues than simply rejecting the entire series. Consequently, another database client cannot see that the transaction is ongoing. It hasn't happened yet at one point in time, while it has happened in full (or nothing) at another.

Database invariants are maintained via consistency, which guarantees that a transaction can only change the database from one consistent state to another. Any data added to the database must be legitimate following all established rules, including constraints, cascades, triggers, and any combination of these. This stops unauthorized transactions from corrupting databases. Referential integrity ensures there is a link between the primary key and the foreign key.

Frequently, transactions are carried out simultaneously (for example, several transactions are reading and writing a table simultaneously). By ensuring that transactions are completed concurrently, isolation keeps the database in the same state that it would have been in had the transactions been executed sequentially. Concurrency control's primary objective is isolation; depending on the level of isolation employed, an incomplete transaction's effects may not be apparent to other transactions.

The durability of transactions ensures that it will stay committed even in the event of a system failure (such as a crash or power loss). This typically indicates that non-volatile memory stores records of completed transactions and their consequences.

There are also four categories of commands:

- DDL (data definition language) is a syntax used to create and edit database objects including users, tables, and indices. DDL statements, which are used to define data structures, particularly database schemas, are comparable to a computer programming language. Typical examples of DDL statements are CREATE, ALTER, and DROP.
- DQL (data query language) can query the data included in schema objects. DQL commands are designed to retrieve the schema relation given the query that is submitted to it.

- DML (data manipulation language) is a programming language for computers that is used to update, remove, and add data to databases. A DML often consists of some of the operators found in wider database language, such as SQL, and is a sublanguage of that language. Some operators may conduct both selecting (reading) and writing. Read-only data selection is sometimes distinguished as being part of a separate data query language (DQL), but it is also closely linked and occasionally regarded as a component of a DML.
- DCL (data control language) supports authorization as the process of controlling access to data contained in a database.

The experiments will focus on the main operations of an application, DQL (SELECT commands to query the content of the database), and DML (INSERT, UPDATE, and DELETE commands).

Java-based applications apply these four operations on the database with the support of different frameworks like JPA, Hibernate, Spring Data JPA, etc. They are responsible for the object-relational mapping. This means that they map a Java model to specific tables from a database.

The MyBatis persistence framework uses an XML descriptor or annotations to link objects with stored procedures or SQL statements. MyBatis maps Java methods to SQL statements rather than Java objects to database tables, in contrast to ORM frameworks. All database features, including views, stored procedures, sophisticated queries, and vendor-specific features, are usable with MyBatis, and for denormalized or legacy databases, or to have complete control over SQL execution, it is frequently a good option. These frameworks have different approaches to handling the mapping problem. Java Database Connectivity (JDBC) is one way to work with databases in Java because it offers APIs that allow the user to execute SQL statements.

Java Database Connectivity (JDBC) is an application programming interface (API) for the Java programming language that defines how a client may access a database. It is a Java-based data access technology used for Java database connectivity. It is part of the Java Standard Edition platform. It provides methods to query and update data in a database and is oriented toward relational databases. A JDBC-to-ODBC bridge enables connections to any ODBC-accessible data source in the Java Virtual Machine (JVM) host.

Since JDBC is mostly a collection of interface definitions and specifications, it allows multiple implementations of these interfaces to exist and be used by the same application at runtime. The API provides a mechanism for dynamically loading the correct Java packages and registering them with the JDBC Driver Manager (DriverManager). DriverManager is used as a connection factory for creating JDBC connection environments.

The Java programming language's application programming interface (API), known as Java Database Connectivity (JDBC), specifies how a client may access a database. It is a data access technique built on top of Java that is utilized to link Java databases. It is included in the platform of Java Standard Edition. It is focused on relational databases and offers ways to query and change data in a database. Connections to any ODBC-accessible data source in the Java Virtual Machine (JVM) host are made possible using a JDBC-to-ODBC bridge.

JDBC permits numerous implementations of these interfaces to exist and be utilized by the same application at runtime because it is primarily a set of interface definitions and standards. The API offers a way to install the appropriate Java packages dynamically and register them with the JDBC Driver Manager (DriverManager). To establish JDBC connections, DriverManager functions as a connection factory.

Object Relational Mapping is a programming method for transferring data between an object-oriented programming language's heap and a relational database. By doing this, a virtual object database that is accessible from within the computer language is effectively created. Data-management operations in object-oriented programming operate on objects that transform scalar values into objects. The programming language treats each of the entries as a single object (it can be referred to using a single variable carrying a reference to the object, for example).

Object Relational Mapping makes the development effort simpler as the JDBC, as it hides the SQL interaction; it offers development with objects instead of database tables; there is no need to take care of the database implementation; there is less code written for the same job; and it is based on the underlying JBDC.

Database connections in Java applications are managed through a JDBC driver, specific to the RDBMS. A specific dialect also needs to be set up by the developers when configuring the database connection so that the queries are created in the right language.

The most popular RDBMSs today are as follows:

- MySQL;
- Oracle SQL;
- Microsoft SQL Server;
- PostgreSQL.

Selecting one of the dialects mentioned above and combining it with the frameworks that best match the requirements can be a hard decision to make. Some combinations might give faster performances for CRUD operations but with the price of writing a lot of code.

This research intends to help developers select the correct combination of technologies that suits their application's needs. For example, an application might not need a fast response from the database, so it should focus on keeping the code simpler and easier to maintain. Also, if speed is crucial for the implementation needs, then the architects and developers must consider a trade-off between code complexity and speed. Furthermore, inefficient combinations will also be discovered in the research. This will help developers avoid bad, unoptimized solutions.

The research would like to assess the behavior of an average-complexity Java application. Joins will be a key element in our application because this is a key operation when working with a relational database.

## 4. Architecture of the Application and Implementation of the Solution

This research aims to help developers choose the best combination of components depending on their use case, by providing actual results using different sizes of records in scenarios.

Each combination will be a separate and independent project, as they must not interfere with each other. This way, a safer evaluation can be conducted. In the future, it can be changed to choose the corresponding framework configuration file and to execute the specific test, because the entities that the solution exposes are used in all the different approaches. Code reuse can be achieved through this method of development, which is a great principle for an application.

All implementations must follow a standard for the data model because, otherwise, the results might become irrelevant. The standard is the Java Persistence API, on top of which all the selected frameworks are mapped.

The solution is based on two main components: the ORM framework and the RDBMS. The responsibility of the ORM framework is mapping Java objects and database models. Also, it controls the flow of data between the application and the database [38].

For testing purposes, the chosen framework was JUnit [39]. This is the most popular testing extension in the Java community. JUnit 5 introduced modular architecture, which brings only the necessary dependencies into the testing code, making it smaller and more efficient. It is constantly being updated by the team that develops it. The tests are easily extendable using third-party extensions, and for this, it was particularly useful to add the Spring extension, in the context of the Spring Data JPA tests.

To improve the testing, Java Microbenchmark Harness (JMH) [40] was added to the project to check if warming-up the Java Virtual Machine will reduce the execution times. It can also be used for extracting code optimization paths or measuring the execution of different methods.

The architecture of the testing application is described in Figure 1, demonstrating how different combinations of the implementation are created:

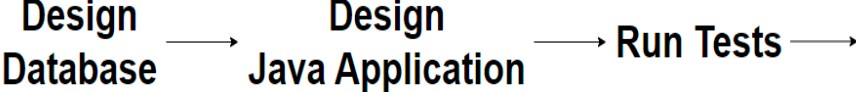

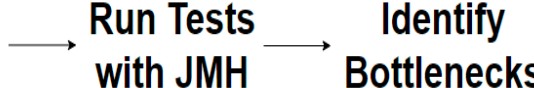

**Figure 1.** Research methodology.

A combination that will be subject to testing is represented by one component from the left side (ORM framework) plus one on the right side (RDBMS), both with and without warm-up. The color legend is as follows:

- BLUE → JPA with all RDBMS;
- GREEN → Hibernate with all RDBMS;
- ORANGE → Spring Data JPA with all RDBMS.

Each testing scenario is mapped with an arrow. This way, all the possibilities of these technologies are covered, and it provides more reliable results. The Java application can be considered the central part because it is the entry point from which the project will start.

The current research expands on our previously published papers, looks for optimization possibilities, and investigates the critical regions.

The application we designed represents a soccer betting service model. The user can create a different number of tickets, each with bets and corresponding matches. The entities of the project are ticket, bet, and match.

Figure 2 presents the data model with the relationship between entities. There are a single-to-many relationships between match and bet. Figure 3 shows the data model in use for testing.

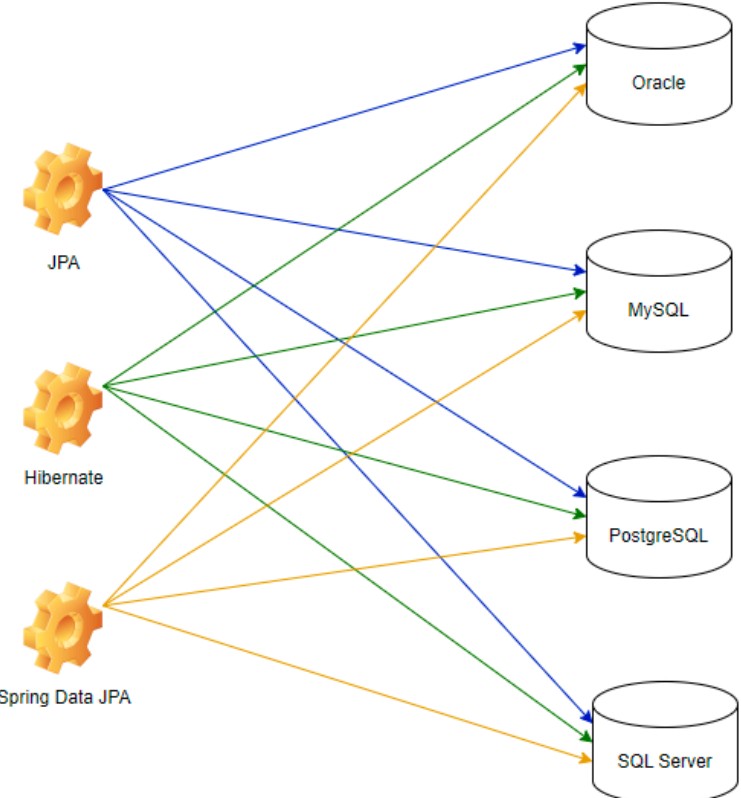

**Figure 2.** Testing architecture including the combinations framework—RDBMS.

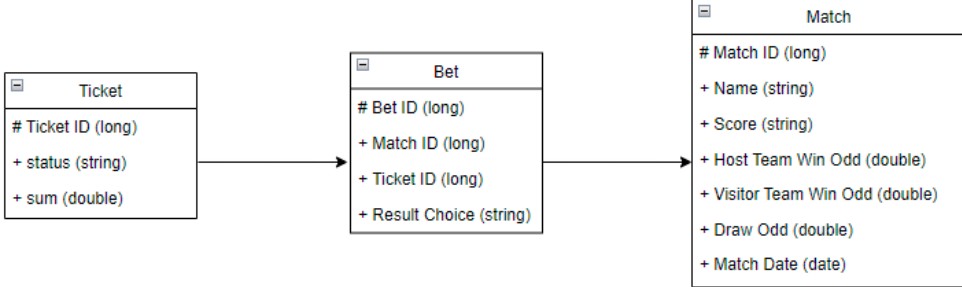

**Figure 3.** The data model for the application to be tested.

The tables have several attributes:

- Ticket.
    - Ticket id -> the unique identifier for a ticket;
    - Status -> one of the following values: WON (if all the bets are successful), LOST (if at least one bet is wrong), PENDING (otherwise);
    - Sum -> represents the amount placed on the ticket as the betting value.
- Bet.
    - Bet id -> the unique identifier for a bet;
    - Match id -> the identifier of the match the bet is placed on;
    - Ticket id -> the identifier of the ticket the bet belongs to;
    - Result choice -> one of the following: 1 (host wins), 2 (visitor wins), X (even).
- Match
    - Match id -> the unique identifier for a match;
    - Name -> a string like "Real Madrid–Barcelona";
    - Score -> the final score of the match;
    - Host team win odd -> the odd for the host's win;
    - Visitor team win odd -> the odd for the visitor's win;
    - Draw odd -> the odd in case the match is even;
    - Match date -> the date when the match will be played.

The setup of the machine where experiments were run is the same as for the previously published work [8]:

- CPU: Intel i7–6700HQ @ 2.6 GHz
- RAM: 8 GB
- Operating System: Windows 10 Pro 64-bit
- Java version: Oracle JDK 17.0.4 x64, JRE (build 17.0.4 + 11)

Differentiating from the point of view of the amount of code to be written for accessing the database, the tests look like this:

- JPA, 36 lines;
- Hibernate, 36 lines;
- Spring Data JPA, 28 lines (a gain of 23%).

The tests intend to measure the execution of each CRUD action while varying the number of tickets as follows: 1000, 2000, 5000, 10,000, 20,000, 50,000, 100,000, 200,000, and 500,000. The updated operation will make multiple changes: the ticket's status, the bet's outcome selection, and the name of the match. There are 12 configuration combinations on which the tests were executed: three ORM frameworks multiplied by four RDBMSs.

Our new experiments involved Java Microbenchmark Harness (JMH), which was responsible for handling the warm-up phase of the Java Virtual Machine. It was set up to only run one iteration for a warm-up and one for measurement, due to the extended running time for larger entry numbers, and both iterations had their results interpreted.

Executing the tests with the same available resources and in the same conditions was another important point that was taken care of in the evaluation.

## 5. Evaluation without JMH

### 5.1. Java Persistence API

We will summarize the results of the previous research to make it clear where the new steps started from. Figures 4–7 provide the results of the execution times without JMH, using Java Persistence API as a framework, and different RDBMSs, as conducted and published in [6–8]. Then, the research will focus on the improvements of using JMH with one warm-up iteration and one measurement iteration (Section 6) and the bottlenecks investigation (Section 7).

The combination MySQL-JPA needs less than 20 min for all four CRUD operations working with 500 records. The READ operation is the fastest, CREATE is a little slower than UPDATE, while DELETE doubles the execution time with the doubling of the records, exhibiting a linear behavior.

For Oracle, the DELETE operation requires almost 2 h on 500,000 entries. The CREATE operation is about 10 times quicker, but still much slower than for MySQL. However, the UPDATE operation behaves better than for MySQL.

Microsoft SQL Server has clearly the best execution times for the UPDATE operation. This advantage is covered by a greater disadvantage: the DELETE operation runs for 4.5 h for 500,000 entries which is a very bad performance when looking at the MySQL test. READ has the best time for this RDBMS for over 100,000 entries, while CREATE fits in the average time so far in the tests.

PostgreSQL is a very good choice when working with a reduced number of records (up to 5000). The performances, however, decrease for a larger number of entries, with remarkably bad behavior for the DELETE operation at 500,000 entries.

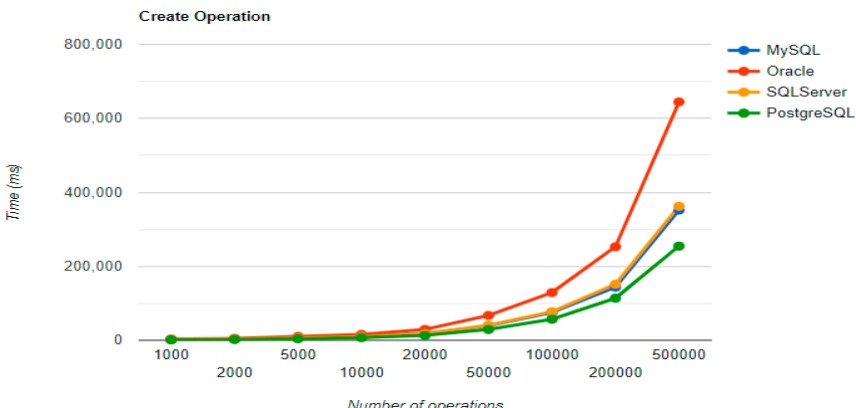

**Figure 4.** Create execution times using JPA without warm-up.

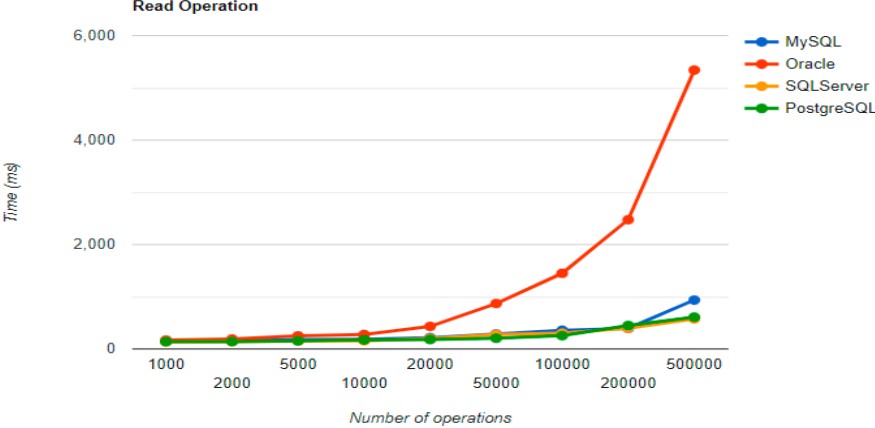

**Figure 5.** Read execution times using JPA without warm-up.

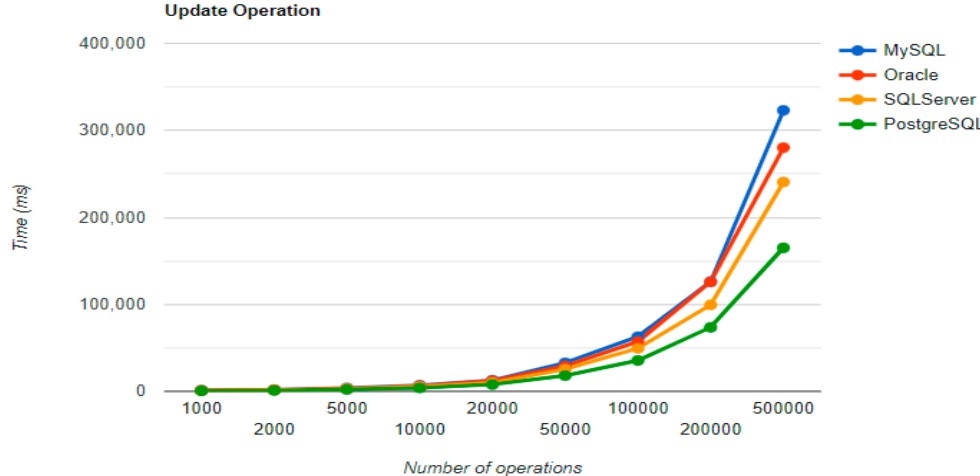

**Figure 6.** Update execution times using JPA without warm-up.

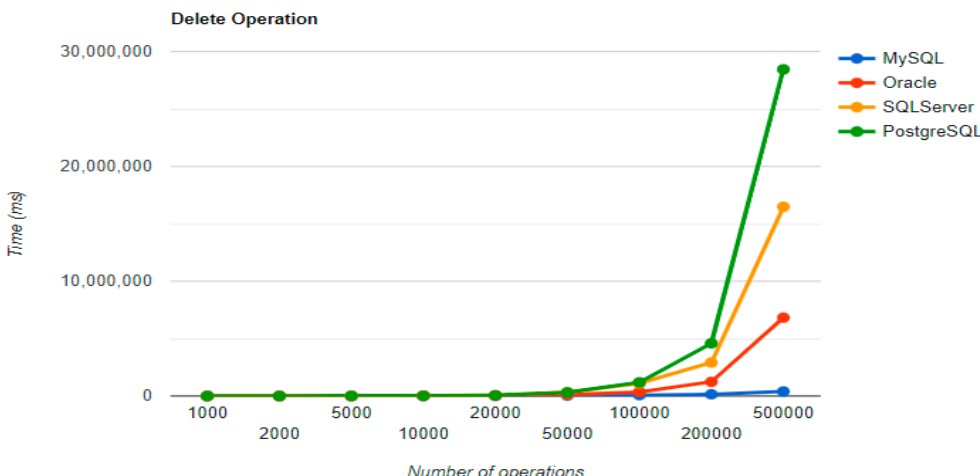

**Figure 7.** Delete execution times using JPA without warm-up.

*5.2. Hibernate*

Figures 8–11 provide the results of the execution times without JMH, using Hibernate as a framework, and different RDBMSs.

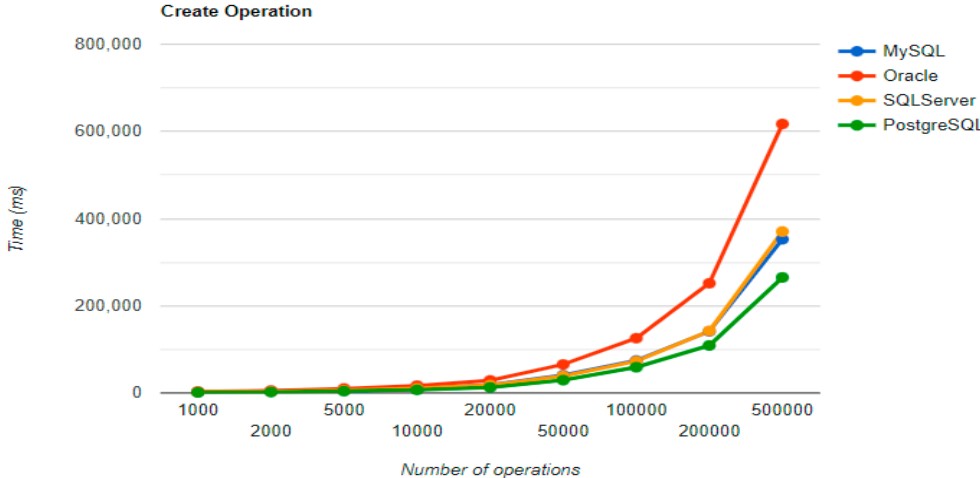

**Figure 8.** Create execution times using Hibernate without warm-up.

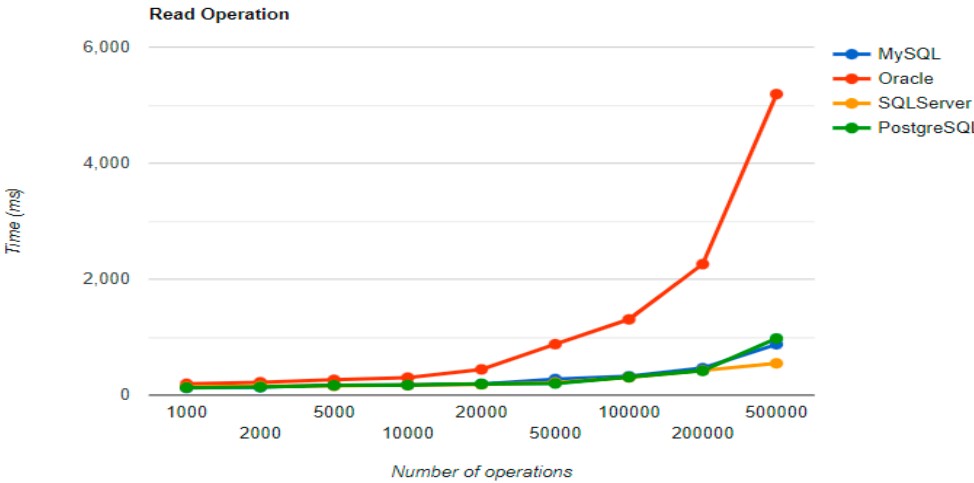

**Figure 9.** Read execution times using Hibernate without warm-up.

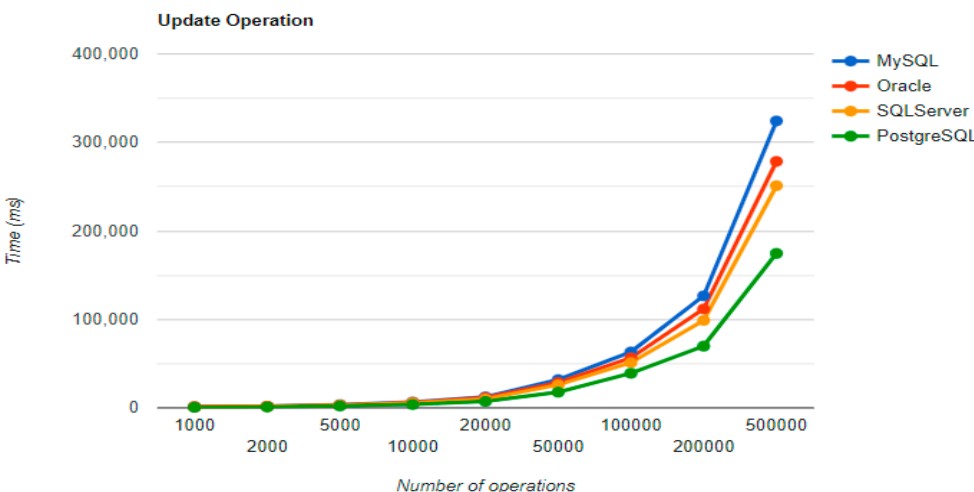

**Figure 10.** Update execution times using Hibernate without warm-up.

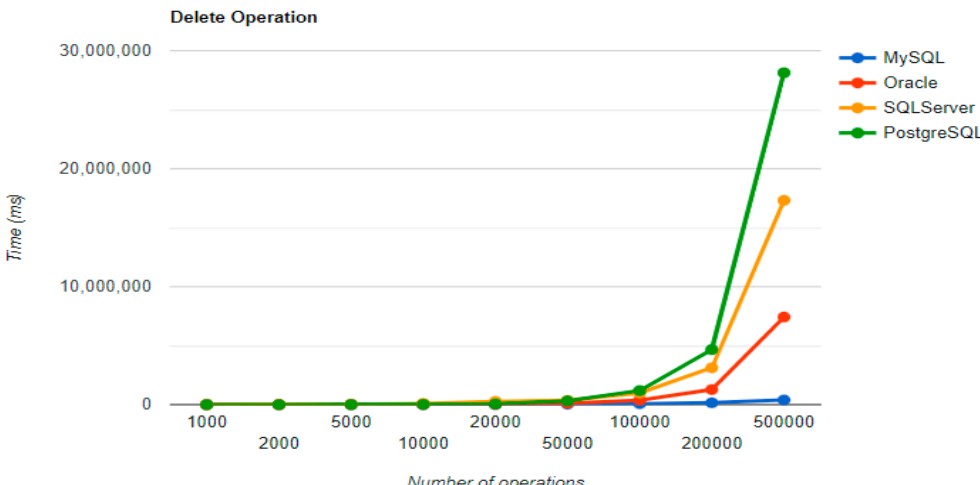

**Figure 11.** Delete execution times using Hibernate without warm-up.

For small data-sets with less than 5000 entries, PostgreSQL is a good option. Increasing the number of records, a different RDBMS may be a better option. When there are 500,000 entries, the deletion performance becomes much slower compared to when there are fewer entries. On the other hand, the advantage of having a larger number of entries is

demonstrated by the improvement in three CRUD operations. The substantial decrease in the deletion performance may make PostgreSQL unsuitable for this scenario.

There are similar insertion and reading times for MySQL, PostgreSQL, and SQL Server, Oracle being the only one that is sluggish from these operations. Despite that, it is the second best option for deleting records, which is the most time-consuming operation.

*5.3. Spring Data JPA*

Figures 12–15 provide the results of the execution times without JMH, using Spring Data JPA as a framework, and different RDBMSs.

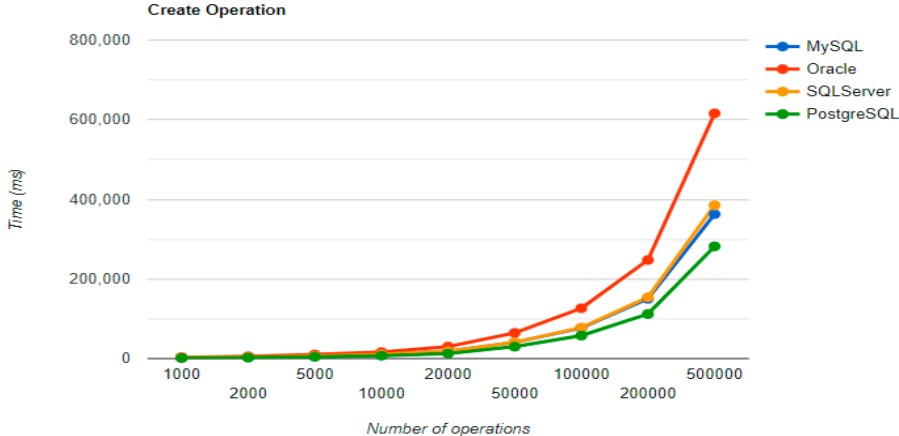

**Figure 12.** Create execution times using Spring Data JPA without warm-up.

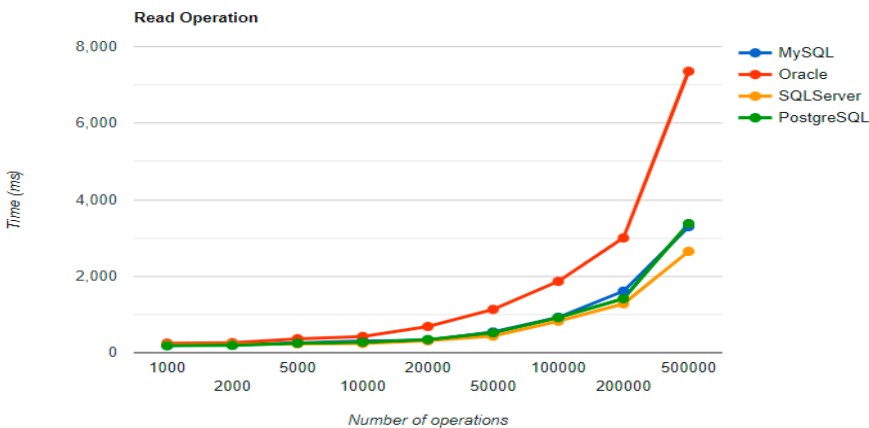

**Figure 13.** Read execution times using Spring Data JPA without warm-up.

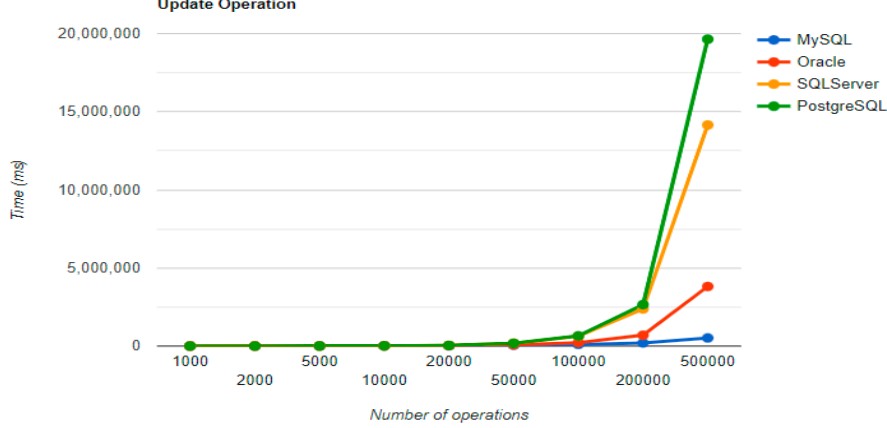

**Figure 14.** Update execution times using Spring Data JPA without warm-up.

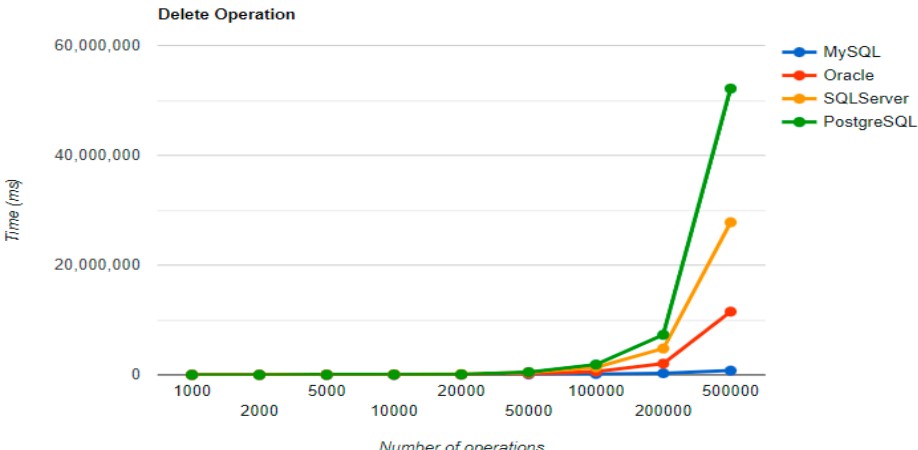

**Figure 15.** Delete execution times using Spring Data JPA without warm-up.

Spring Data JPA introduces its particular overhead while executing the operations. Although the creation time is the shortest of all Oracle implementations for 200k+ entries, the other three operations are much slower, the deletion process takes over 3 h, while the UPDATE operation is less performant compared to the MySQL implementation.

The third RDBMS, Microsoft SQL Server, experiences difficulties with the DELETE and UPDATE operations. The deletion of 500k entries takes almost 8 h, and the update needs about half of this time. The creation is like the MySQL combination.

## 6. Evaluation with JMH

Section 7 summarized the results of the previous experiments [6–8]. We focus now on the improvements by introducing Java Microbenchmark Harness (JMH) in the measurements. The benchmark was set to run with one warm-up iteration, followed by one measurement iteration, after the warm-up. The execution times will be presented for both iterations, as it could offer an interesting interpretation of the results.

### 6.1. Java Persistence API, Warm-Up Iteration

Figures 16–19 provide the results of the execution times of the warm-up iteration, with JMH, using Java Persistence API as a framework, and different RDBMSs.

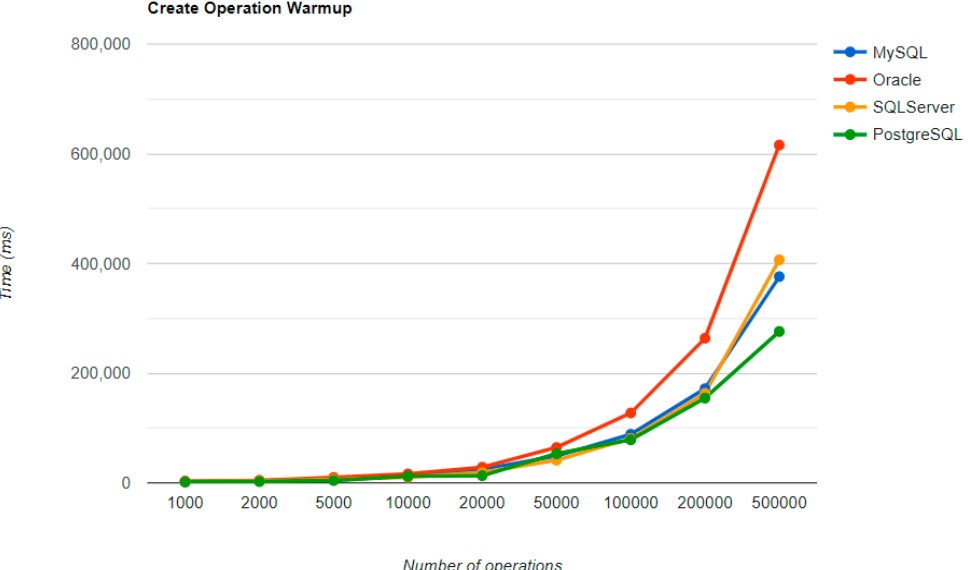

**Figure 16.** Create warm-up iteration execution times using JPA.

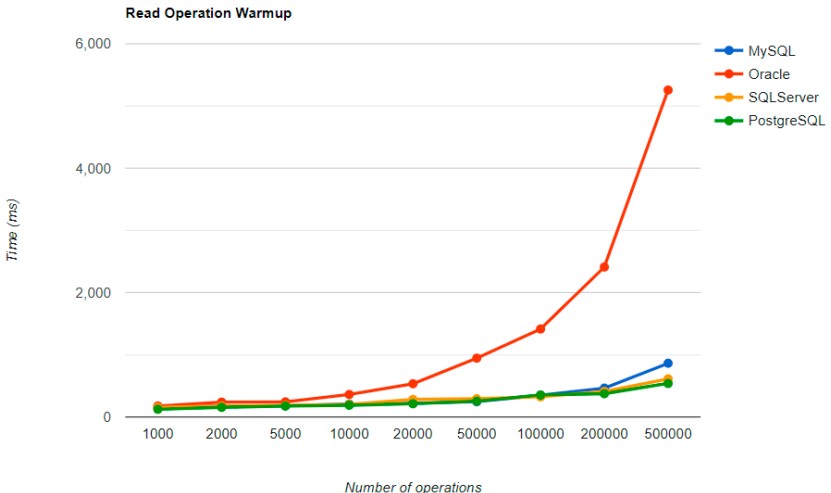

**Figure 17.** Read warm-up iteration execution times using JPA.

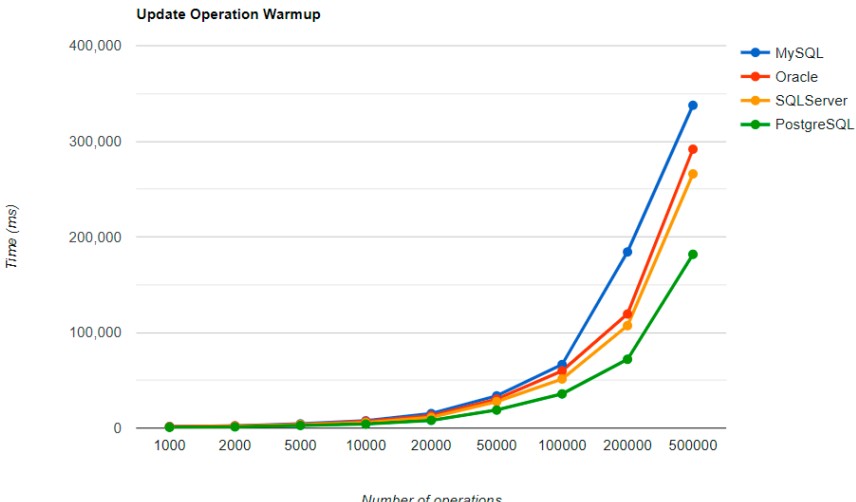

**Figure 18.** Update warm-up iteration execution times using JPA.

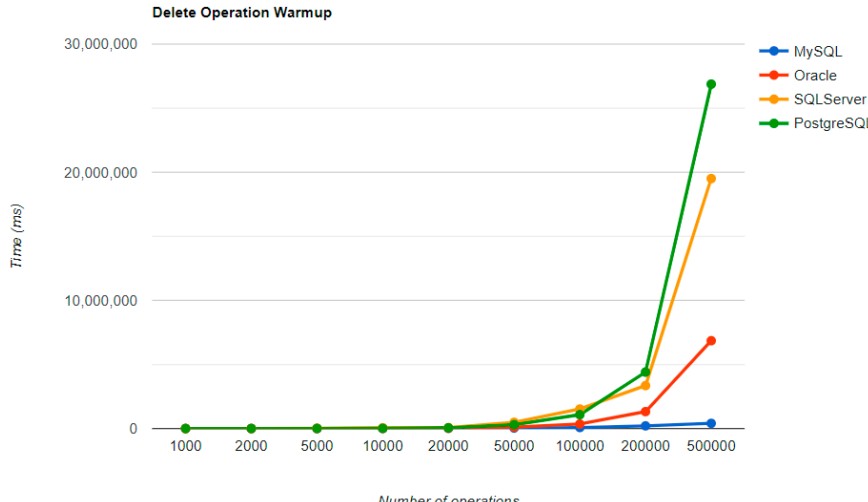

**Figure 19.** Delete warm-up iteration execution times using JPA.

*6.2. Hibernate, Warm-Up Iteration*

Figures 20–23 provide the results of the execution times of the warm-up iteration, with JMH, using Hibernate as a framework, and different RDBMSs.

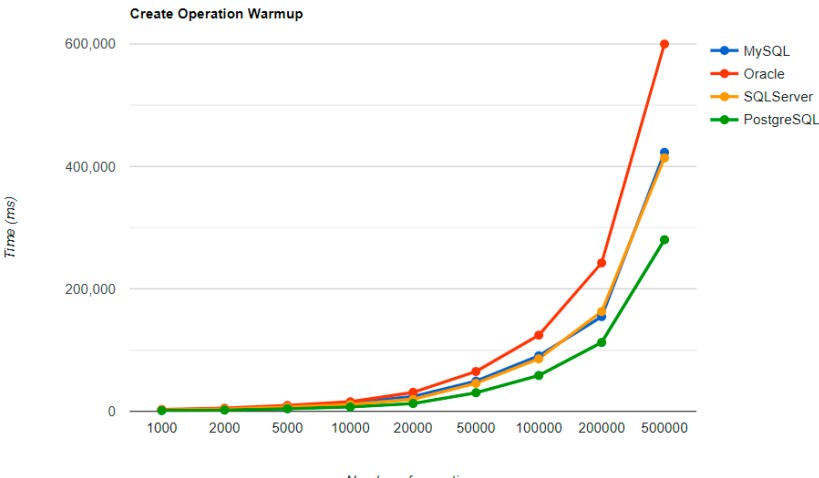

**Figure 20.** Create warm-up iteration execution times using Hibernate.

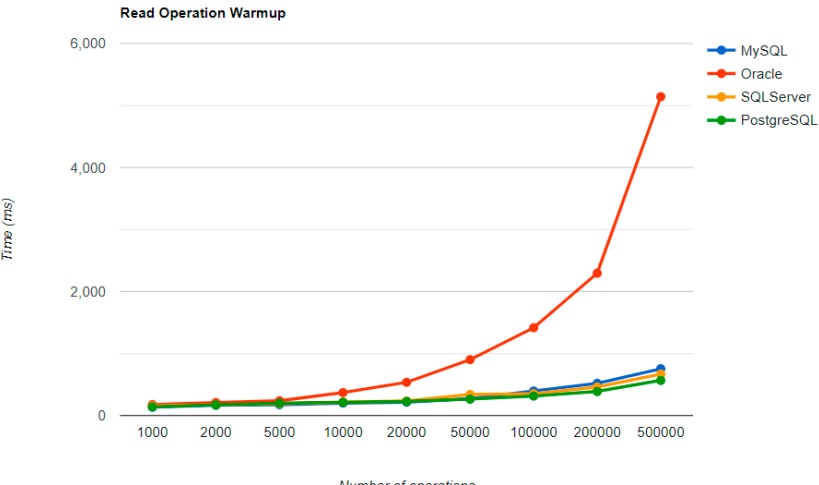

**Figure 21.** Read warm-up iteration execution times using Hibernate.

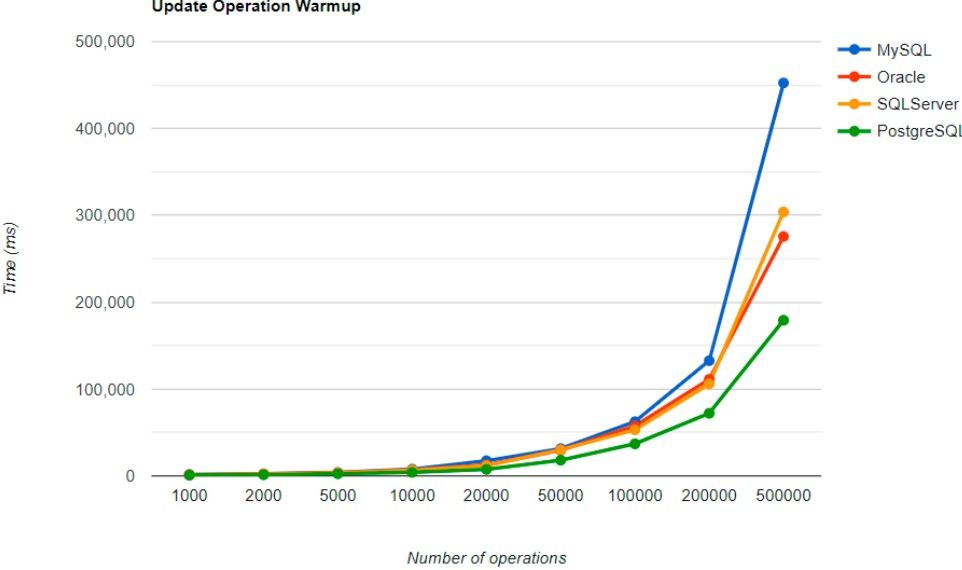

**Figure 22.** Update warm-up iteration execution times using Hibernate.

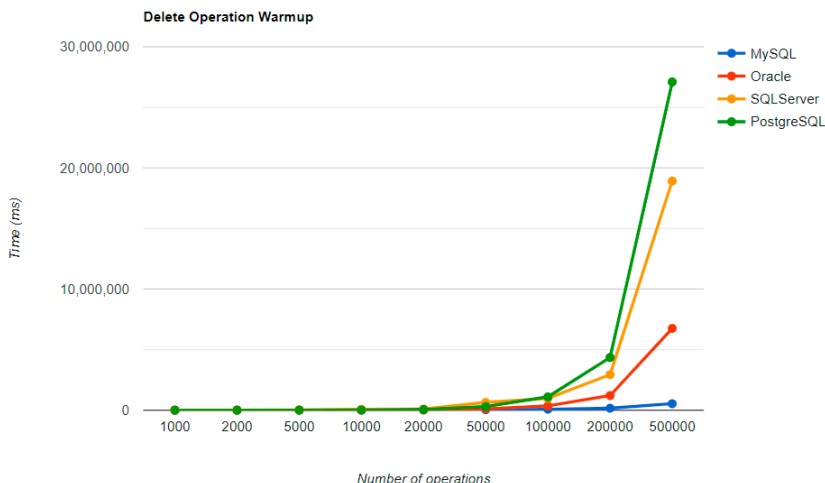

**Figure 23.** Delete warm-up iteration execution times using Hibernate.

### 6.3. Spring Data JPA, Warm-Up Iteration

Figures 24–27 provide the results of the execution times of the warm-up iteration, with JMH, using Spring Data JPA as a framework, and different RDBMSs.

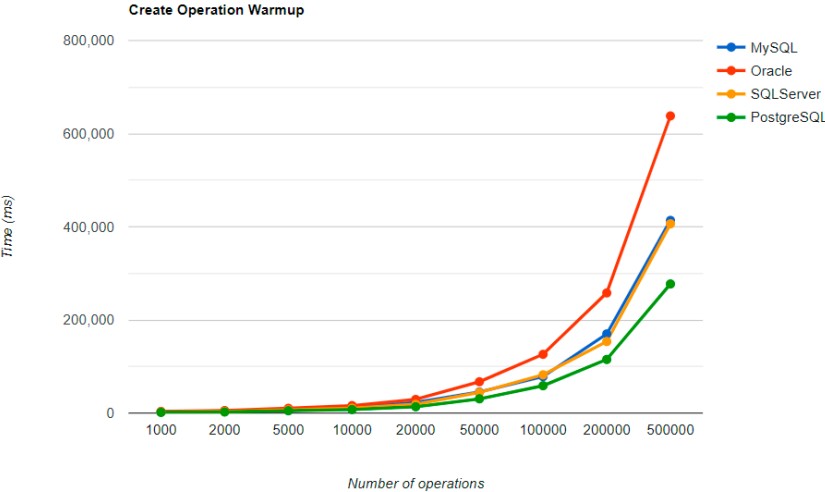

**Figure 24.** Create warm-up iteration execution times using Spring Data JPA.

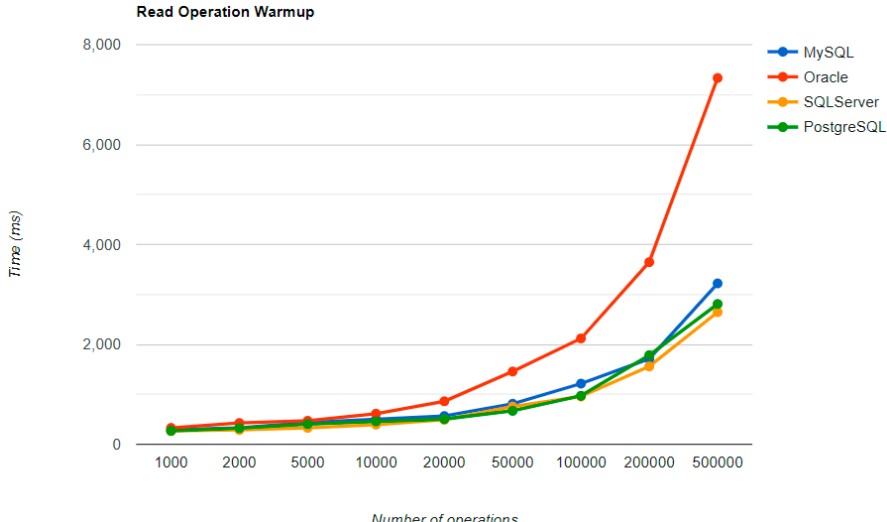

**Figure 25.** Read warm-up iteration execution times using Spring Data JPA.

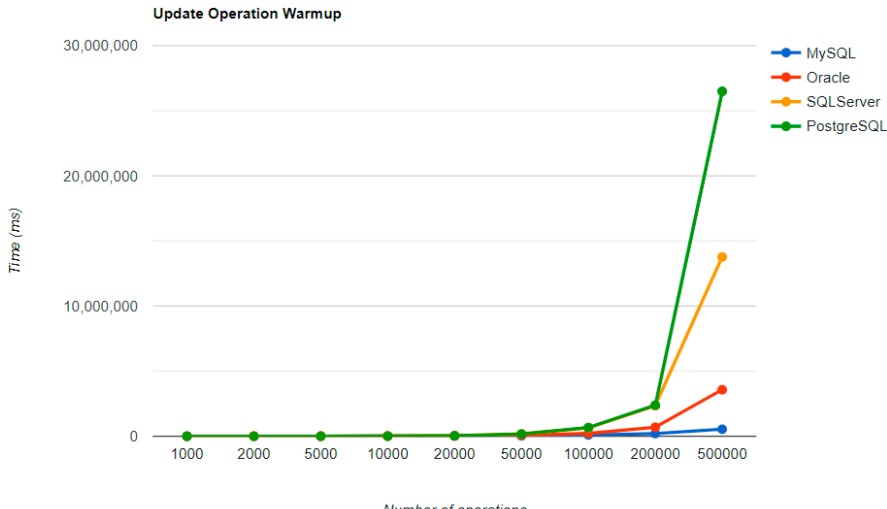

**Figure 26.** Update warm-up iteration execution times using Spring Data JPA.

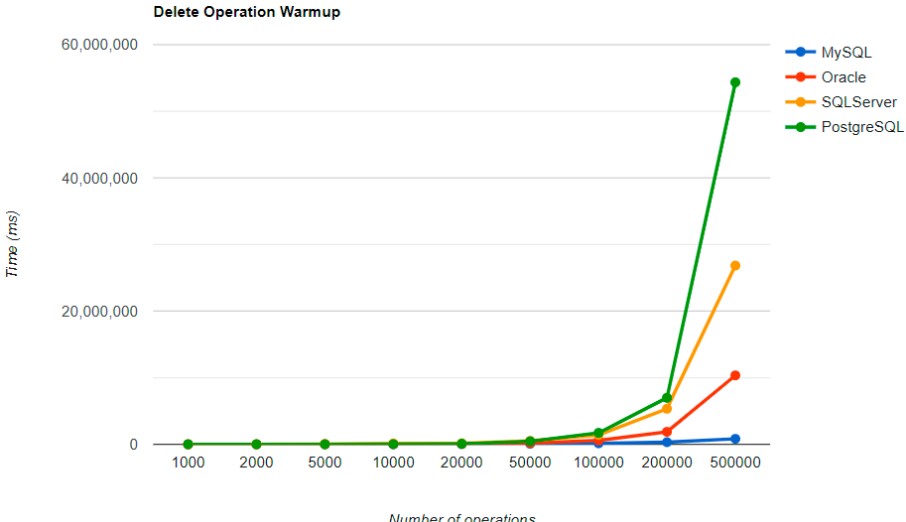

**Figure 27.** Delete warm-up iteration execution times using Spring Data JPA.

MySQL provides the best DELETE time. The UPDATE operation has the worst execution time in the warm-up.

The second RDBMS tested with this framework, Oracle, has the worst READ execution time by far, and the warm-up does not change its position at all. The same is true for the CREATE operation. A good aspect of Oracle is the deletion time, as it is placed in second place, after MySQL.

SQL Server has some visible changes after the warm-up. At first, it has a pretty good time on the CREATE operation for fewer entries than 100k, but it becomes the worst at this operation for 500k entries or more. It improves after warming-up at the READ operation, jumping from the second position to the first one. Something that remains the same is the DELETE operation, the second worst one. The best RDBMS for creating, reading, and updating without a warm-up is PostgreSQL, which has an issue with the slowest deletion. Creating and updating are the only positions maintained after the warm-up phase.

### 6.4. Java Persistence API after Warm-Up

Figures 28–31 provide the results of the execution times after the warm-up iteration, with JMH, using Java Persistence API as a framework, and different RDBMSs.

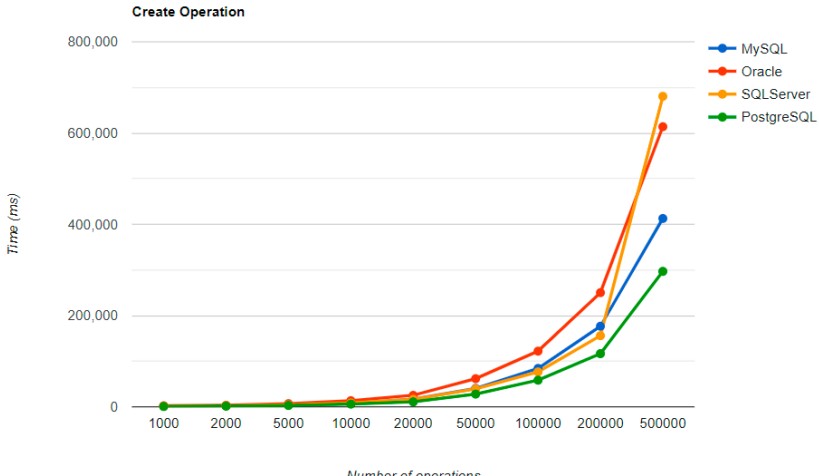

**Figure 28.** Create execution times using JPA, after warm-up.

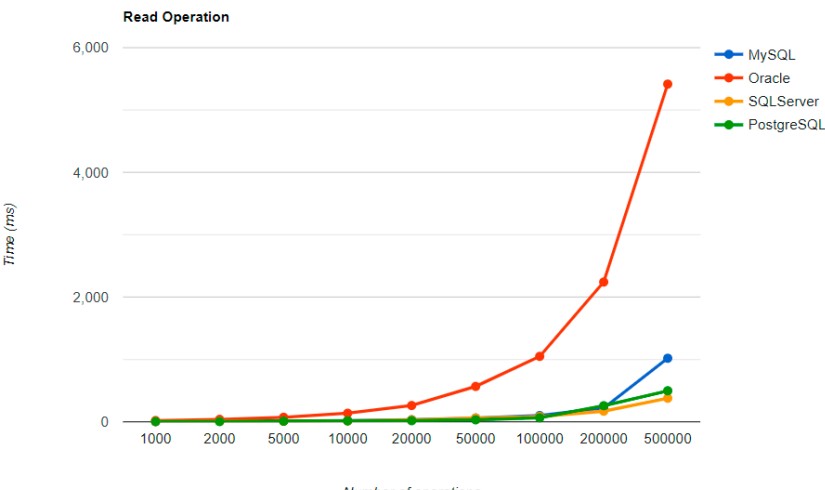

**Figure 29.** Read execution times using JPA, after warm-up.

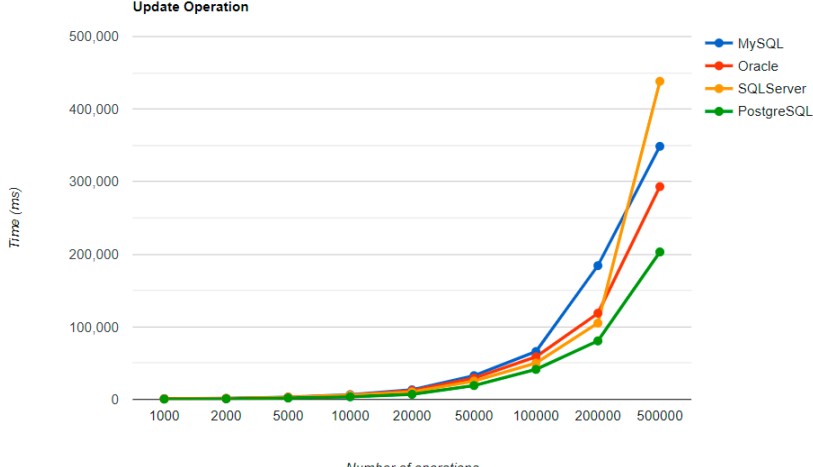

**Figure 30.** Update execution times using JPA, after warm-up.

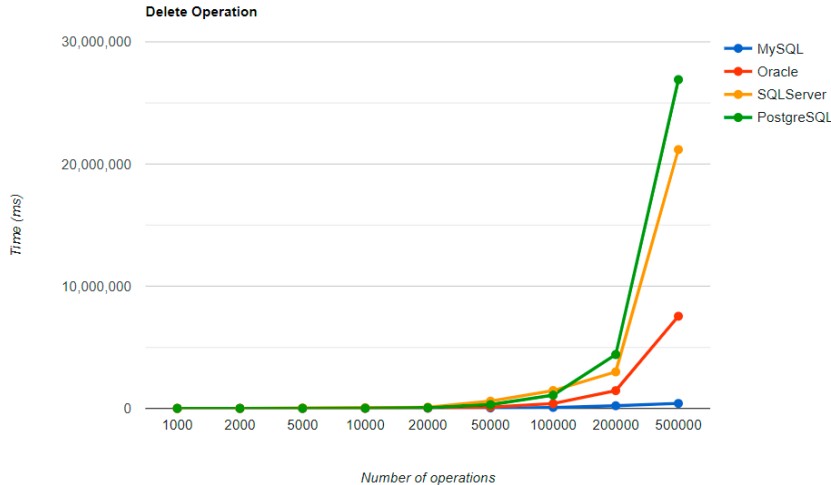

**Figure 31.** Delete execution times using JPA, after warm-up.

### 6.5. Hibernate after Warm-Up

Figures 32–35 provide the results of the execution times after the warm-up iteration, with JMH, using Hibernate as a framework, and different RDBMSs.

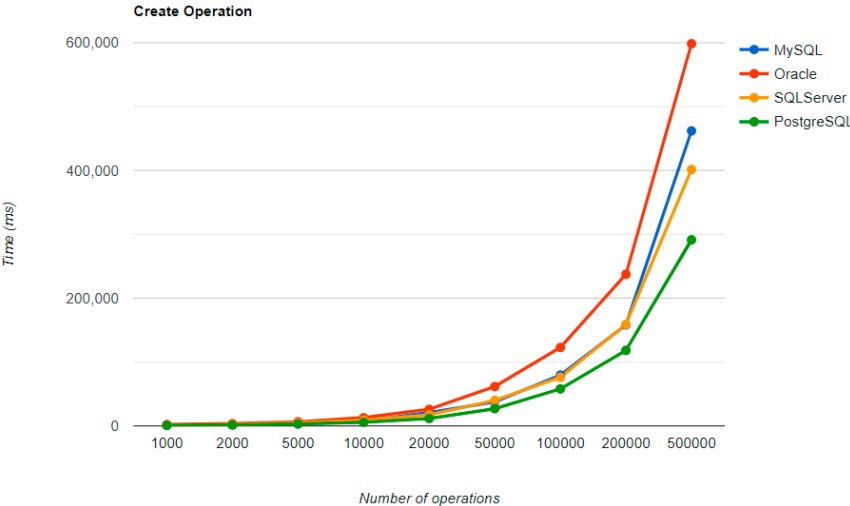

**Figure 32.** Create execution times using Hibernate, after warm-up.

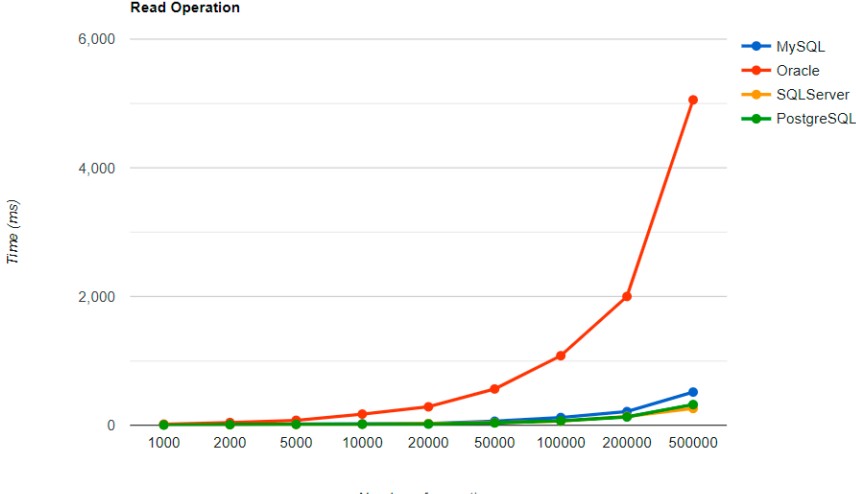

**Figure 33.** Read execution times using Hibernate, after warm-up.

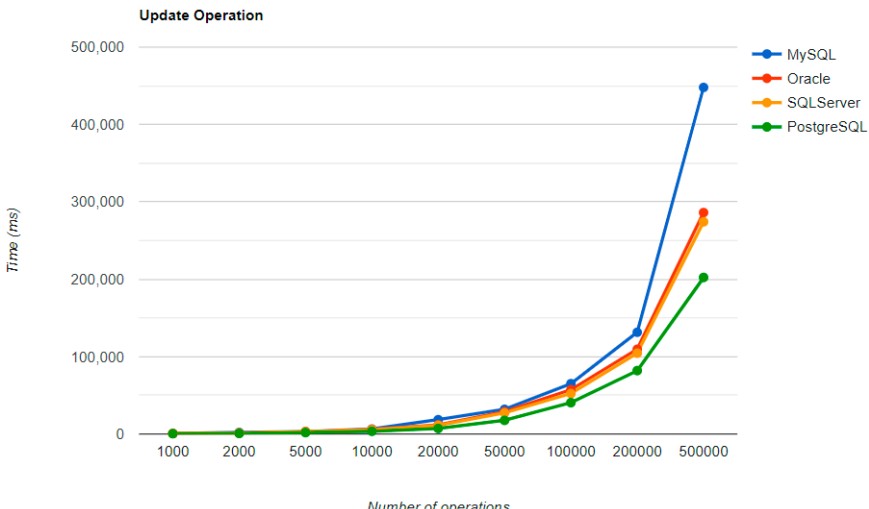

**Figure 34.** Update execution times using Hibernate, after warm-up.

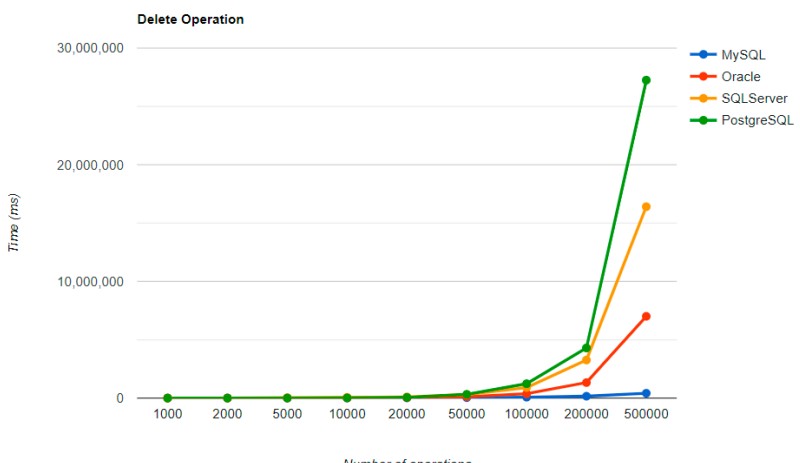

**Figure 35.** Delete execution times using Hibernate, after warm-up.

Regarding the warm-up execution times, PostgreSQL has the best timing on the CREATE and UPDATE operations for each number of entries tested, but it is terrible when it comes to deleting more than 100k entries.

Reading is also great with PostgreSQL, but this time for over 50k entries. For less than this number, MySQL seems to be the best choice. The most valuable point for MySQL is deletion, which is more than 12.5 times faster than the second RDBMS's (Oracle), for a 500k entry run.

Oracle reads a bit sluggishly, especially for the READ operation, almost seven times slower at the greatest run than MySQL, which is placed in the third position. Its strengths are updating and deleting for big numbers of entries, making it the second-best choice for these operations. Oracle is the worst solution for less than 50k entries, at every operation.

Ranking second for the CREATE and READ operations, SQL Server is a solution to be taken into consideration when developing a new application, but only if it is not planned to make lots of deletions, because it performs almost as badly as PostgreSQL for this operation.

The most visible impact the warm-up had on the second run is observed when looking at the READ operation. It runs almost instantly for a number of entries smaller than 50k and reaches a maximum of half a second for 500k entries, except for the Oracle RDBMS, but a big improvement can be noticed here also.

Even if the execution times after the warm-up iteration are expected to be lower, this is true only for the CREATE operation measured on the Oracle and SQL Server RDBMSs.

This operation has better results on MySQL and PostgreSQL too, but not for more than 100k entries.

On the MySQL RDBMS, the UPDATE and DELETE operations have almost the same performance, with or without warm-up. The same thing is available for deleting on an SQL Server RDBMS or a PostgreSQL one.

Updating execution times are reduced for less than 50k entries with MySQL, 100k entries with PostgreSQL, and 500k entries with Oracle. The most visible change overall was for Oracle, which reduced durations for almost every number of entries.

The final results after the second iteration made some slight changes in comparison to the performance for each operation, with each RDBMS. However, PostgreSQL still has the best timing for creating and updating entries, and also the worst overall timing when it comes to deletions.

SQL Server comes forward with the best reading, over-passing PostgreSQL and MySQL, which falls to the third position after it was first in the warm-up phase. Oracle provides some serious time reduction in less than 200k entries, but at the largest run, it has almost the same time as before, a bit over 5 s.

The best deletion is taken again by MySQL, with an even bigger difference from the warm-up comparison: almost 17 times faster than the Oracle performance. But, this RDBMS has poor performance when it comes to updating entries because it is the worst solution of all four.

### 6.6. Spring Data JPA after Warm-Up

Figures 36–39 provide the results of the execution times after the warm-up iteration, with JMH, using Spring Data JPA as a framework, and different RDBMSs.

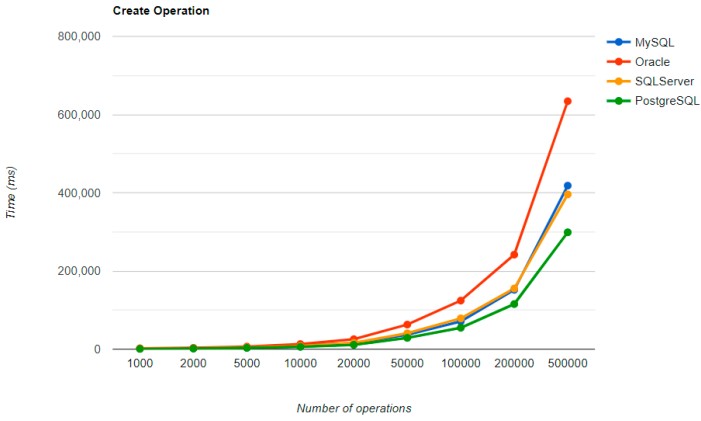

**Figure 36.** Create execution times using Spring Data JPA, after warm-up.

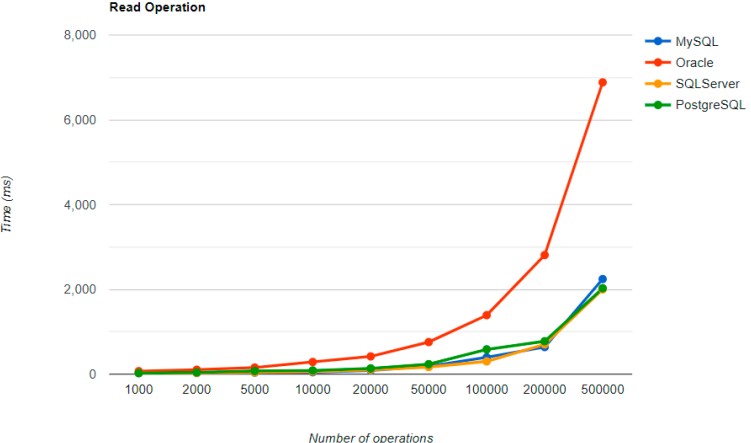

**Figure 37.** Read execution times using Spring Data JPA, after warm-up.

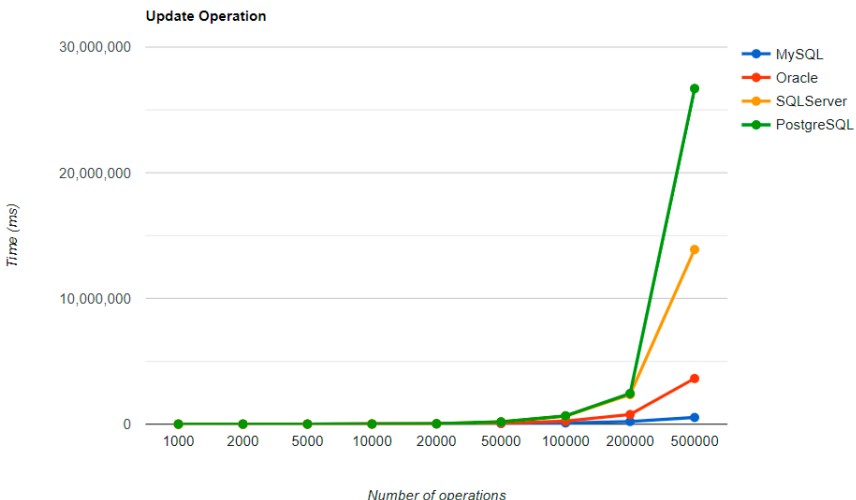

**Figure 38.** Update execution times using Spring Data JPA, after warm-up.

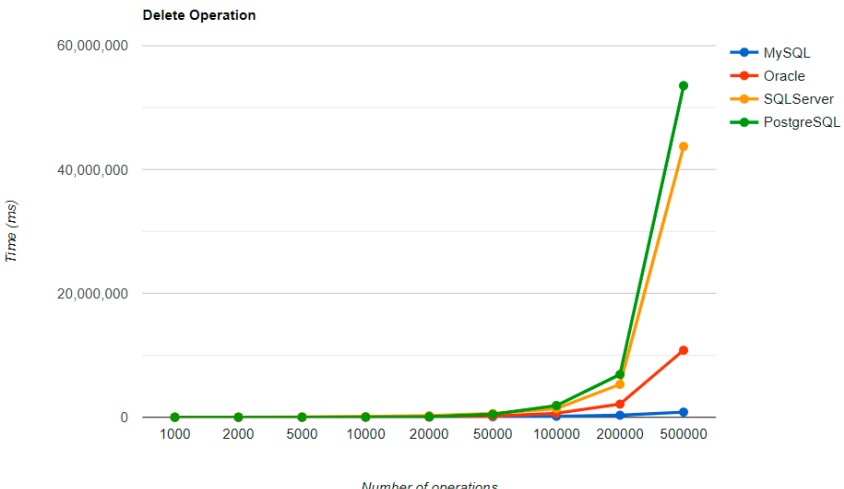

**Figure 39.** Delete execution times using Spring Data JPA, after warm-up.

MySQL has the fastest performance for both UPDATE and DELETE operations, while its other CRUD operations have an average performance like other RDBMS tested using the benchmark.

Oracle has the slowest READ time, and this does not change with a warm-up iteration. It also performs poorly in CREATE operations. However, Oracle has the second fastest time for both DELETE and UPDATE.

SQL Server maintains its position after warm-up, unlike in the JPA implementation. It has the fastest READ performance, but the second slowest DELETE and UPDATE, after PostgreSQL. PostgreSQL has the best performance for CREATE operations, but the slowest for DELETE and UPDATE, making it the least ideal choice for this framework. These statistics remain unchanged after the warm-up iteration.

Compared to the other two ORM frameworks, Spring Data JPA brings a big overhead to the table, making it interesting for further investigations at the level of the internal operations that slow down the execution. An advantage of this framework is the reduced size of the code that is to be written for the database interaction, and a trade-off between the development speed and the execution times needs to be considered.

## 7. Bottlenecks Investigation

A quick comparison between this research study and the previous one presented in [5] revealed some common findings:

- Spring Data JPA has a big overhead, the slowest of all frameworks for batch operations;
- Hibernate and JPA solutions go side by side, with almost overlapping graphs;
- Spring Data JPA requires the fewest lines of code.

The execution times in milliseconds for 50k entries for both solutions for each framework and operation on MySQL database are presented below in Table 1.

**Table 1.** Results for both solutions for 50k entries on MySQL.

| Operation | Current Research | | | Previous Research | | |
|---|---|---|---|---|---|---|
| | JPA | Hibernate | Spring Data JPA | JPA | Hibernate | Spring Data JPA |
| Create | 39,153 | 40,688 | 40,862 | 16,463 | 16,512 | 59,629 |
| Read | 287 | 278 | 542 | 344 | 362 | 2252 |
| Update | 32,834 | 31,887 | 51,424 | 16,355 | 16,276 | 75,071 |
| Delete | 39,249 | 40,001 | 73,632 | 12,768 | 12,857 | 79,799 |

For JPA and Hibernate, the execution times are quite linear for each CRUD operation except reading, which is reasonable considering that the current research works with three entities and the previous one works with a single entity.

Reading seems to be slower on a single database table, but this could be due to a newer more powerful generation of Intel CPU that has been used in the current research.

Both these approaches revealed a big overhead in the execution times for Spring Data JPA for each operation and RDBMS tested. This triggered an interest in investigating the time discrepancies between the mentioned framework and the other ones (Hibernate and JPA). The overhead for Spring Data JPA is even bigger for the previous research, despite having just one entity, which makes the research on the bottlenecks more interesting.

### 7.1. Spring Data JPA Bottlenecks Investigation

The experiments demonstrate that Spring Data JPA comes with the least amount of code written, but also with a big overhead in the execution times, in comparison with JPA and Hibernate.

To locate the bottlenecks, an analysis of the framework methods execution times had to be conducted. This was achieved using YourKit Java Profiler 2022.9. Alongside information about methods and times, it can extract call trees or lists of hot spots, thus facilitating the identification of performance bottlenecks [41].

Previous research indicated that more than half of performance bottlenecks originate in the data access layer [42]. The current analysis intended to locate more accurately the bottlenecks and to retrieve the top five hot spots for each RDBMS for runs with the number of entries starting from 50k and comparing them with the top five hot spots (% of running time) for a 50k run with Hibernate.

The potential differences found after this comparison could explain the overhead observed for Spring Data JPA. These differences would then be investigated to see if there is something extra that the Spring solution verifies.

The percentage of time spent on the most time-consuming methods identified using YourKit Java Profiler is presented in the Figures 40–51, using RDBMS, framework, and the number of entries (100% means the execution time of the test).

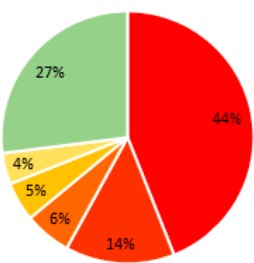

MySQL – Hibernate hot spots for 50k entries

- com.mysql.cj.jdbc.ClientPreparedStatement.executeUpdate() – 44%
- org.hibernate.internal.SessionImpl.persist(Object) – 14%
- com.mysql.cj.jdbc.ConnectionImpl.prepareStatement(String, int) – 6%
- com.mysql.cj.jdbc.ConnectionImpl.prepareStatement(String) – 5%
- java.util.Properties.load(InputSteam) – 4%
- Time spent in other methods - 27%

**Figure 40.** MySQL—Hibernate hot spots for 50k entries.

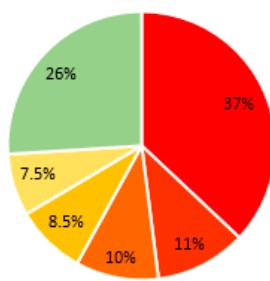

MySQL – Spring Data JPA hot spots for 50k entries

- com.mysql.cj.jdbc.ClientPreparedStatement.executeUpdate() – 37%
- org.hibernate.internal.SessionImpl.persist(Object) – 11%
- com.mysql.cj.jdbc.ClientPreparedStatement.executeQuery() – 10%
- com.mysql.cj.jdbc.ConnectionImpl.prepareStatement(String) – 8.5%
- com.mysql.cj.jdbc.ConnectionImpl.prepareStatement(String, int) – 7.5%
- Time spent in other methods - 26%

**Figure 41.** MySQL—Spring Data JPA hot spots for 50k entries.

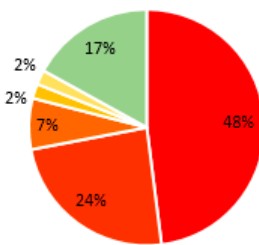

MySQL – Spring Data JPA hot spots for 100k, 200k, and 500k entries

- com.mysql.cj.jdbc.ClientPreparedStatement.executeUpdate() – 48%
- com.mysql.cj.jdbc.ClientPreparedStatement.executeQuery() – 24%
- com.mysql.cj.jdbc.ConnectionImpl.prepareStatement(String) – 7%
- com.mysql.cj.jdbc.ConnectionImpl.prepareStatement(String, int) – 2%
- org.hibernate.internal.SessionImpl.persist(Object) – 2%
- Time spent in other methods - 17%

**Figure 42.** MySQL—Spring Data JPA hot spots for 100k, 200k, and 500k entries.

Oracle – Hibernate hot spots for 50k entries

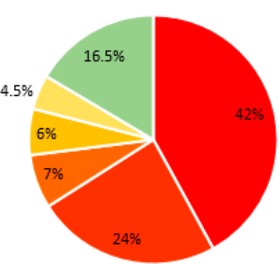

■ oracle.jdbc.driver.OraclePreparedStatementWrapper.executeUpdate() – 42%

■ org.hibernate.internal.SessionImpl.persist(Object) – 24%

■ oracle.jdbc.driver.PhysicalConnection.prepareStatement(String, String[]) – 7%

■ oracle.jdbc.driver.PhysicalConnection.prepareStatement(String) – 6%

■ java.util.Properties.load(InputSteam) – 4.5%

■ Time spent in other methods - 16.5%

**Figure 43.** Oracle—Hibernate hot spots for 50k entries.

Oracle – Spring Data JPA hot spots for 50k entries

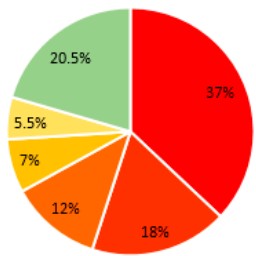

■ oracle.jdbc.driver.OraclePreparedStatementWrapper.executeUpdate() – 37%

■ org.hibernate.internal.SessionImpl.persist(Object) – 18%

■ oracle.jdbc.driver.OraclePreparedStatementWrapper.executeQuery() – 12%

■ oracle.jdbc.driver.PhysicalConnection.prepareStatement(String, String[]) – 7%

■ oracle.jdbc.driver.PhysicalConnection.prepareStatement(String) – 5.5%

■ Time spent in other methods - 20.5%

**Figure 44.** Oracle—Spring Data JPA hot spots for 50k entries.

Oracle – Spring Data JPA hot spots for 100k, 200k, and 500k entries

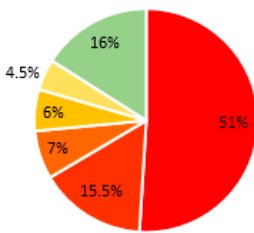

■ org.postgresql.jdbc.PgPreparedStatement.executeUpdate() – 51%

■ org.hibernate.internal.SessionImpl.persist(Object) – 15.5%

■ org.postgresql.jdbc.PgConnection.prepareStatement(String, int) – 7%

■ org.postgresql.jdbc.PgConnection.prepareStatement(String) – 6%

■ java.util.Properties.load(InputSteam) – 4.5%

■ Time spent in other methods - 16%

**Figure 45.** Oracle—Spring Data JPA hot spots for 100k, 200k, and 500k entries.

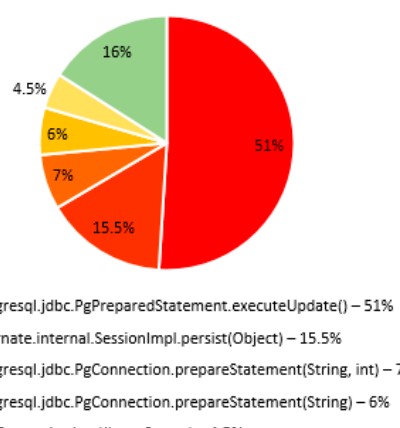

**Figure 46.** PostgreSQL—Hibernate hot spots for 50k entries.

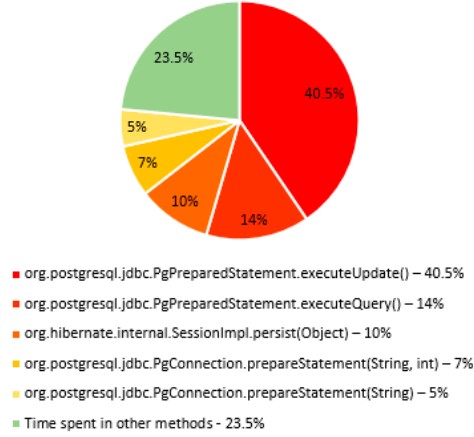

**Figure 47.** PostgreSQL—Spring Data JPA hot spots for 50k entries.

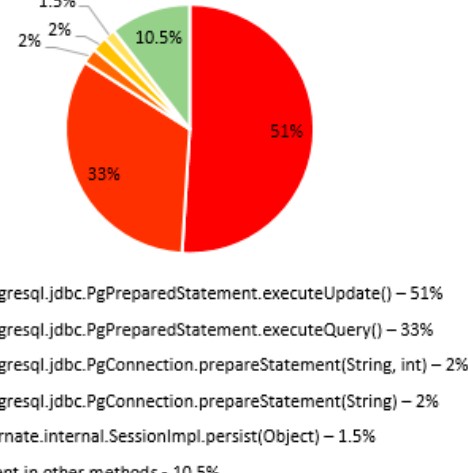

**Figure 48.** PostgreSQL—Spring Data JPA hot spots for 100k, 200k, and 500k entries.

SQLServer – Hibernate hot spots for 50k entries

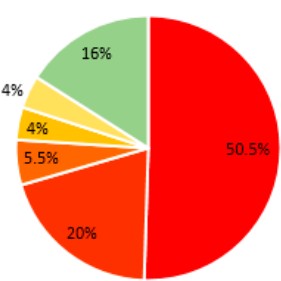

■ com.microsoft.sqlserver.jdbc.SQLServerPreparedStatement.executeUpdate() – 50.5%

■ org.hibernate.internal.SessionImpl.persist(Object) – 20%

■ com.microsoft.sqlserver.jdbc.SQLServerConnection.prepareStatement(String, int) – 5.5%

■ com.microsoft.sqlserver.jdbc.SQLServerConnection.prepareStatement(String) – 4%

■ java.util.Properties.load(InputSteam) – 4%

■ Time spent in other methods - 16%

**Figure 49.** SQLServer—Hibernate hot spots for 50k entries.

SQLServer – Spring Data JPA hot spots for 50k entries

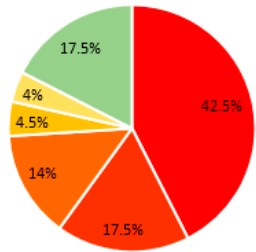

■ com.microsoft.sqlserver.jdbc.SQLServerPreparedStatement.executeUpdate() - 42.5%

■ com.microsoft.sqlserver.jdbc.SQLServerPreparedStatement.executeQuery() – 17.5%

■ org.hibernate.internal.SessionImpl.persist(Object) – 14%

■ com.microsoft.sqlserver.jdbc.SQLServerConnection.prepareStatement(String, int) – 4.5%

■ com.microsoft.sqlserver.jdbc.SQLServerConnection.prepareStatement(String) – 4%

■ Time spent in other methods - 17.5%

**Figure 50.** SQLServer—Spring Data JPA hot spots for 50k entries.

SQLServer – Spring Data JPA hot spots for 100k, 200k, and 500k entries

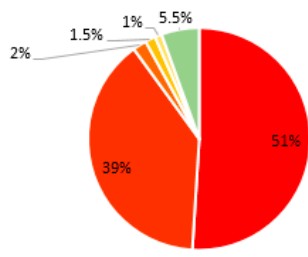

■ com.microsoft.sqlserver.jdbc.SQLServerPreparedStatement.executeUpdate() – 51%

■ com.microsoft.sqlserver.jdbc.SQLServerPreparedStatement.executeQuery() – 39%

■ com.microsoft.sqlserver.jdbc.SQLServerConnection.prepareStatement(String, int) – 2%

■ com.microsoft.sqlserver.jdbc.SQLServerConnection.prepareStatement(String) – 1.5%

■ org.hibernate.internal.SessionImpl.persist(Object) – 1%

■ Time spent in other methods - 5.5%

**Figure 51.** SQLServer—Spring Data JPA hot spots for 100k, 200k, and 500k entries.

The main thing that can be noticed from the results above is that Spring Data JPA makes really time-consuming calls with the executeQuery() method, which is missing in the

Hibernate implementations. That happens because when Spring Data JPA deletes an entry, at first it looks for the entry and then it makes the deletion. The deletions are completed one by one with the deleteAll(Iterable<Ticket>). Since the delete operation is the most time-consuming one, the executeQuery() method becomes one of the most active methods.

Another interesting fact is that most of the time is spent inside drivers' methods for each RDBMS tested and not in a framework method.

*7.2. Reducing Spring Data JPA Bottlenecks*

The first attempt to reduce the bottlenecks was to override the void delete(Ticket) method from the created SimpleJpaRepository<Ticket, ID>. To do this, we created an interface that extends CrudRepository<Ticket, ID>:

```
@NoRepositoryBean
public interface CustomCrudRepository<ID> extends CrudRepository<Ticket, ID> {
    void delete(Ticket entity);
}
```

An implementation of this interface was needed to override the method and skip the queries given to the database:

```
public class CustomJpaRepository<ID> extends SimpleJpaRepository<Ticket, ID>
                                implements CustomCrudRepository<ID> {
    private EntityManager entityManager;

    @Override
    public void delete(Ticket entity) {
        this.entityManager.remove(this.entityManager.contains(entity) ?
                            entity : this.entityManager.merge(entity));
    }
}
```

After creating the repository bean and injecting the EntityManager dependency into it, profiling was performed again. There was an improvement of about 10% in the execution times, but it was still not close to the other two ORM frameworks tested.

There were also interesting pop-up messages while the investigation with YourKit finished for Spring Data JPA runs: "Potential deadlock: Frozen threads found -> frozen for at least 19 min 10 s". This happened for a run with PostgreSQL and Spring Data JPA with 100k entries, but also for runs with other RDBMSs. The record was established with a PostgreSQL run with Spring Data JPA with 500k entries: 1 d 0 h 47 min 18 s, which explains why the biggest scenario runs for so long.

The problem is not a deadlock, but a long starvation, as the program finishes its run successfully. Because of that, another attempt to reduce the bottlenecks was to replace the transaction management completed by Spring Data JPA.

Removing the @EnableTransactionManagement annotation was the first step in the process. Then, the EntityManagerFactory used for the JPA implementation was used to replace the factory used by Spring Data JPA.

The CustomCrudRepository<ID> interface was also modified. The logic for all CRUD operations did not change too much; they were overridden to use custom transaction management through the EntityManagerFactory. After these changes took place, the

execution times for 50k entries for each CRUD operation for each RDBMS tested with Spring Data JPA were remeasured and the results are presented in Tables 2 and 3.

**Table 2.** Execution times in milliseconds, 50k entries, after transaction management refactor.

| Operation | Relational Database Management System | | | |
|---|---|---|---|---|
| | **MySQL** | **Oracle** | **SQLServer** | **PostgreSQL** |
| Create | 43,371 | 68,977 | 42,253 | 31,839 |
| Read | 314 | 976 | 261 | 271 |
| Update | 33,955 | 29,537 | 27,668 | 18,924 |
| Delete | 42,988 | 113,404 | 381,440 | 326,854 |

**Table 3.** Execution times in milliseconds for 50k entries with Hibernate.

| Operation | Relational Database Management System | | | |
|---|---|---|---|---|
| | **MySQL** | **Oracle** | **SQLServer** | **PostgreSQL** |
| Create | 40,688 | 65,406 | 39,076 | 29,389 |
| Read | 278 | 878 | 218 | 202 |
| Update | 31,887 | 28,601 | 25,955 | 17,682 |
| Delete | 40,001 | 112,778 | 359,862 | 314,171 |

Comparing the new results with the Hibernate ones proves a significant decrease in time for Spring Data JPA when the transaction management is changed.

It emphasizes that the purpose of changing Spring Data JPA's transaction management is experimental, with the purpose of investigating how the classic EntityManager behaves in Spring's environment. It does not intend to solve the bottlenecks; further research on transaction management needs to be conducted.

Some of the previous papers provided experiments focusing on particular RDBMSs or on improving access to the databases using optimizer hints, optimization of the execution plans, prediction, diagnosis, and tuning approaches. Other previous work focused on contemporary challenges such as the work with Big Data, Cloud Computing, or using machine learning to obtain better performances.

Our research differentiates itself from the previously published ones with the extensive combination of four types of databases and three ORM frameworks. Alongside this, we investigated the critical execution sections (bottlenecks) and the improvements that the JVM warm-up brings.

## 8. Conclusions

To emphasize the performance differences after introducing the warm-up, we will briefly review the conclusions of the experiments that were not using it.

For the work without JMH, A great overall performance is offered by MySQL. It has incredibly good execution times, despite having a slower performance than PostgreSQL for creating, reading, and updating entries, but it saves a lot of time on the delete operation.

The experiments were designed to provide multi-criteria analysis, using different RDBMSs together with different ORM frameworks. This way, software engineers may decide based on the specificity of their projects and on the operations that they forecast to be intensive for their circumstances.

Performance is similar for Hibernate and Java Persistence API, while Spring Data JPA brings a lot of overhead with it, but it also offers an easier solution regarding the code dimension to access and modify data.

After the warm-up phase, the only CRUD operation that had a visible and constant improvement was the READ one. For the rest of the operations, the improvements are

generally noticed on a small number of entries (less than 20k or 50k). What may be surprising is that sometimes, usually for a large number of entries (500k), the warm-up phase is useless, as the execution times for the second iteration are even bigger.

After researching what could make Spring Data JPA's transaction management act so slow, the conclusion was that it does not have just an EntityManager, but it has a SharedEntityManagerCreator which creates more objects of type EntityManager to avoid possible thread safety issues.

The classic EntityManager generated using EntityManagerFactory used in the JPA implementation does not offer thread safety, but in the proposed experiment this is not necessary. Using it makes Spring Data JPA faster in its interaction with the relational database management system.

Switching the transaction management turns Spring Data JPA into a similar solution in terms of performance like JPA or Hibernate. This means that the starvation situations mentioned by the YourKit profiler are happening somewhere inside the transaction mechanism proposed by Spring Data JPA.

Designing enterprise applications nowadays is a real challenge and involves a lot of high-level skills and experience. Alongside designing [43] and assessing the architecture [44], selecting and applying the software development methodology [45], and testing the functionality [39], performance plays an essential role in modern software. Selecting the best database and ORM that suits a project is a matter of finding a trade-off between the execution speed on one side, and development speed on the other side, and extensive experimental results strongly support such a decision.

In future work to continue the research, we consider extending the experiments for larger databases exceeding 500k rows; extending the research for NoSQL databases as MongoDB; evaluating the performance impact of connection pooling and other methods of database optimization; investigating the possible solutions for the bottlenecks we identified; investigate if the bottlenecks change in Spring Data JPA with a hierarchical object model instead of the flat relational model used; analyze if the performance of CRUD operations changes with different database schemas, including more indexes, constraints, and triggers; compare performance for batch CRUD operations; and analyze the performances of other frameworks like EclipseLink and MyBatis.

**Author Contributions:** Conceptualization, C.T.; formal analysis, A.M.B. and C.T.; investigation, A.M.B. and C.T.; methodology, A.M.B. and C.T.; software, A.M.B.; supervision, C.T.; validation, A.M.B. and C.T.; writing—original draft, A.M.B.; writing—review and editing, C.T. All authors have read and agreed to the published version of the manuscript.

**Funding:** This research received no external funding.

**Institutional Review Board Statement:** Not applicable.

**Informed Consent Statement:** Not Applicable.

**Data Availability Statement:** The data presented in this study are openly available in GitHub at https://github.com/alexbtn/master_project (accessed on 5 February 2024).

**Conflicts of Interest:** Cătălin Tudose was employed by the company Luxoft Romania. The remaining author declare that the research was conducted in the absence of any commercial or financial relationships that could be construed as a potential conflict of interest.

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
