# Peer review of "Performance Analysis and Improvement for CRUD Operations in Relational Databases from Java Programs Using JPA, Hibernate, Spring Data JPA"

_applsci, doi:10.3390/app14072743_

Round 1
Reviewer 1 Report (Previous Reviewer 2)
Comments and Suggestions for Authors
This paper aims to assist developers in selecting the correct technology combination for their application needs by combining ORM frameworks with RDBMS. Detailed experimental validation and conclusion analysis have been conducted. However, there are still some issues within the paper that need to be addressed:
1. The arrangement of chapters is somewhat disorganized, with each chapter containing relatively little content. It is suggested to optimize the structure of the article.
2. It is recommended to describe the research method steps in Chapter 2 with a flowchart for a more intuitive presentation.
3. Contributions should be elaborated on in points, categorized as theoretical, methodological, and application contributions.
4. Figures 11 and 87 are mislabeled.
5. There are many formatting issues throughout the manuscript, and mistakes in the reference citation format, such as lines 161 and 168. Please carefully revise the format of the article.
6. When citing reference 10 at line 144 of the article, it is recommended that the author do not copy the text from the reference, but generalize the related content with this manuscript and provide an analysis of these viewpoints.
7. When selecting the most suitable database and ORM, is there a standard to give developers a reference?
8. In lines 101-106 of the article, it is suggested to change the bullet point format to paragraph format.
Comments on the Quality of English LanguageSee Comments for Authors
Author Response
Please see the attachment.

Reviewer 2 Report (New Reviewer)
Comments and Suggestions for Authors
Topic is actual, text quality is bellow average, whole article is very long with many very simular tables, graphs. It will be a lot fo work for normalising this article.
abstract - add some numerical results
Intruduction
- its very short, need to be extended
- add some more citatations
- Chapter 2 should be part of introduction
Related works
- this chapter is actualy named "previous research" rename it to "related works"
- this chapter should be after introduction, before background
- at least 10 (20 will be nice) simular works, each work one paragraph
Problem background
- its very short for article with 46 pages, extend it at least to 3-4 pages
Chapter 6
- use more sentences, there are many unorded lists and almost none explanation text
Chapter 7 and 8
- there is too many tables, graphs, you should remove most of them. They look very simular
- try to identify most intersting results and only theres results should have tables and graphs.
Chapter 9
- too many source codes, remove most of them, you can keep only the most interesting
Discussion
- this chapter is missing, you should compare your results with results from related works
Conclussion
- should be not diveded into subchapters
- add idea of future works
References
- too few references
- add at least 20 more refences indexed in WoS
Author Response
Please see the attachment.

Reviewer 3 Report (New Reviewer)
Comments and Suggestions for Authors
1. The manuscript lacks comprehensive references for the techniques and applications discussed. Please, revise the paper thoroughly.
2. The tables and figures occupies a significant portion of the manuscript without offering substantial insights or implications. A more concise presentation that highlights key findings and their relevance would be more effective.
3. The inclusion of both tables and figures to present the same data is redundant and does not add value to the manuscript. We recommend choosing the most effective way to present each data set to avoid repetition and conserve space.
4. The figures presented in the manuscript are challenging to compare, which hinders the reader's ability to understand the results fully. Improving the clarity and format of these figures to facilitate direct comparison is necessary.
5. While quoting sources can be appropriate, excessive quoting can detract from the originality of your manuscript. We encourage the paraphrasing of existing literature to integrate it more seamlessly into your analysis and discussion.
6. The manuscript does not clearly explain the acquisition of hotspot data presented on pages 39 and 40, and these sections lack captions. Providing a detailed methodology and proper labeling for all data is crucial for the reader's understanding.
7. Section 9.2 introduces a method to address specific bottlenecks, but there is insufficient background on the rationale and development of this approach. Expanding on the theoretical and empirical foundations that led to this method would significantly strengthen the manuscript.
Author Response
Please see the attachment.

Reviewer 4 Report (New Reviewer)
Comments and Suggestions for Authors
Suggеstions for improvеmеnts:
11) In thе conclusions and highlight 1 2 kеy practical guidеlinеs that еmеrgе from thе rеsults to hеlp еnginееrs pick thе bеst tеchnologiеs
22) Thе bottlеnеcks analysis for Spring Data JPA providеs vеry usеful insights. Considеr еxpanding this to divе dееpеr into what causеs thе ovеrhеad comparеd to JPA and Hibеrnatе
33) Invеstigatе if thе bottlеnеcks improvе in Spring Data JPA with a morе rеal world hiеrarchical objеct modеl instеad of thе flat rеlational modеl usеd
44) Analyzе if thе rеlativе pеrformancе of CRUD opеrations changеs with diffеrеnt databasе schеmas е.g. morе indеxеs and constraints and triggеrs
55) Comparе pеrformancе for batch CRUD opеrations instеad of row by row opеrations which arе lеss common
66) Considеr tеsting with largеr databasеs еxcееding 500K rows closеr to production sizеs
77) Evaluatе pеrformancе impact of connеction pooling and othеr databasе optimizations
88) For complеtеnеss and analyzе othеr popular ORM framеworks likе EclipsеLink and MyBatis еtc.
99) Includе morе dеtails on thе databasе sеrvеr hardwarе and OS and JVM vеrsions usеd in thе еxpеrimеntal sеtup
110) Add a small samplе databasе schеma diagram for rеfеrеncе
Author Response
Please see the attachment.

Reviewer 5 Report (New Reviewer)
Comments and Suggestions for Authors
The proposed research is aimed to help developers choose the best combination of components depending on their use case by providing actual results in different sizes of the records scenarios for the combination of several ORM frameworks and several databases. So, the research has a more practical character than scientific one.
According to this, there are some comments:
1) In practice, it is difficult to predict which database operations will be performed predominantly, so the question arises of how necessary it is to research each CRUD operation separately.
2) The volume of the paper can be significantly reduced by showing the results of the study using drawings; tables duplicate the results
3) Conclusions are too long and repeat the summary in sections 7-8
4) However, the overview of the state of the issue is very short
5) Conclusions need to be shortened, making them more generalizable
6) Maybe, the scientific result can be found by summarizing the experiments performed.
Round 2
Reviewer 1 Report (Previous Reviewer 2)
Comments and Suggestions for Authors
The authors have answered all my questions and have revised the manuscript according to the comments.
Comments on the Quality of English LanguageSome minor language editing errors still exists, which may need careful check for that.
Author Response
Thanks for the review.
For the indicated minor language errors, our reviews and the help of Grammarly seemed to be limited. We think that the proof reading will help.
Reviewer 2 Report (New Reviewer)
Comments and Suggestions for Authors
The authors resolved most of my comments and I agree with explanation why some of them were not resolved. I agree with the current version of the article.
Author Response
Thanks for the review and for agreeing with the current version of the article.
Reviewer 3 Report (New Reviewer)
Comments and Suggestions for Authors
If you insist to put the table in the paper, then put them at the end of the paper in the form of appendix.
Also put the figures in a 2 by 2 array, please (Position the legend within the graph area.) Use larger fonts for the graph.
Create | Read |
Update | Delete |
Author Response
Thanks for your notes.
The tables with the results were already separated as additional material. The article contains now only 3 tables, by the end, that are directly connected with the text and should remain here.
Collapsing the figures in a matrix will make them hardly readable. The length of the article is appropriate now, so we think it will be much better if things remain this way.
Reviewer 5 Report (New Reviewer)
Comments and Suggestions for Authors
1) It's possible to recommend to the authors to more specifically explain the essence of the experimental results highlighted in red and yellow
2) These points need somehow arranged either as drawings or in some other way in a generally accepted manner
3) It is necessary to present in a more compact form Fig. 4-7, 8-11, 12-15,16-19, 20-23, 24-27,28-31,32-35, 36-39
Author Response
Thanks for the review.
- We added additional comments to explain the essence of the experimental results highlighted in red and yellow (lines 686-688).
- We replaced the previous tables with pie-charts (lines 689-719).
- Compacting these figures will make them unreadable, so we decided to keep things this way.
This manuscript is a resubmission of an earlier submission. The following is a list of the peer review reports and author responses from that submission.
Round 1
Reviewer 1 Report
Comments and Suggestions for Authors
The work presented in this paper concerns the performance analysis of four popular relational databases when operated through three different Java-based frameworks. The authors consider all possible combinations of databases and frameworks and test create/read/update/delete operations, using or not the possibility to warm up the JVM before query execution.
Although the topic of the article is interesting, there are two very serious concerns: (1) It is not clear how this work extends the previous publications, and (2) There is insufficient explanation of the produced results in the article.
In more detail:
- In Section 4, the authors present some studies concerning similar research. However, it is not made clear how their work differentiates from previous research.
- The architecture presented in Section 5 misses important details making the authors choices sound groundless. For example, JUnit is the chosen framework. What are the reasons of this choice other that the popularity of the framework? What are the characteristics of this framework making it a good candidate? Why are the assertions used for comparing the results? How are they used? What are the expected results?
Also, Figure 1 misses to include the scenario with the JVM warm up.
-The schema given in Section 6 does not capture adequately the mini-world. For example, one ticket can have multi bets. If some of the bets is won and another bet is lost, what will be the status of the ticket?
- It is not clear which part of the work has been previously published by the authors and what is the contribution of this work. The reader is further confused when, in Section 7.1, the authors declare that they will summarize results from previous research but in what follows they never explain when the summarization of the previous results is over.
- Although there are several tables and figures showing the experimental results, there are not comments, explanations, argumentations concerning all these measurements, making the result look unfounded.
Overall, this work is not mature to be published as a journal article.
Comments on the Quality of English LanguageThe main problem is that authors extensively use quotations throughout the article.
Reviewer 2 Report
Comments and Suggestions for Authors
The paper aims to assist developers in selecting the right combination of technologies suited for their application needs by combining ORM frameworks with RDBMS. There is detailed experimental verification and conclusion analysis. However, there are several issues that need to be further improved. Some issues are very key, like a large number of grammar mistakes and format errors, ambiguity of the methodology, which may lead to that it is not applicable for publication in the journal. Some issues in the paper that need to be addressed are as follows:
1. The discussion in Chapter 1 about the background of the problem, the importance of the research field, and the research contributions is not detailed enough.
2. Chapter 2 lacks descriptive information. It is recommended to provide sufficient and detailed information, including a thorough explanation of the research design, experimental process, and data processing methods.
3. The analysis of related work in Chapter 4 is inadequate. It is suggested to analyze the existing related to your own work.
4. There are many formatting issues in the manuscript, such as lines 194 to 208 on page 5, pages 35 to 36, and issues with the formatting of reference citations.
5. There exist many issues with grammar and non-standard use of words. Please revise carefully throughout the whole manuscript.
6. In the experimental chapter, after conducting experiments, there is a lack of detailed analysis, such as the results that can be obtained and explanations for the reasons behind these results.
7. The conclusion section may be redundant and lengthy. And it is inappropriate that the article heavily cites literature and previous work. It is suggested to write a summary of research findings, significance, contributions, and future prospects.
8. The discussion regarding the amount of code in the article is minimal. It is recommended that the author adds a table of the amount of code for different technology combinations to more intuitively show the differences between them.
9. It is advised to include more introductions to previous work, such as Reference 2.
10. In line 543 of the article, Spring Data JPA makes a very time-consuming call to the `executeQuery()` method. It is suggested to use a more intuitive way to show the differences between hotspots of various technologies.
Comments on the Quality of English LanguageSee the comments for authors
Reviewer 3 Report
Comments and Suggestions for Authors
The question analyzed within the paper is of much interest to the scientific and engineering community. Database operations occur in almost every research study, and improvement in the performance of any CRUD operations might save a significant amount of time.
There are questions about the methodology implemented in the paper. The goal of the study needs to be clarified. If the goal
Need to be stated:
-
It must state which Java version and JDK / JRE (Oracle, OpenJDK, Zulu, etc.).
-
Data schemes should be specified, for example, the types of fields in the database. For example, in MySQL, each database has at least two popular engines: InnoDB and Aria. Also, essential test operation modes - single-threaded or multi-threaded could be specified.
-
The attempts to perform some optimizations for one ORM without specifying possible consequences of their application; why is it not done similarly for other ORMs?
-
Profiling - the ratio of CPU time spent within each method should be specified (e.g., when the code inside persist() takes 5% and nested executeUpdate() 95% differs from when the division is 50% and 50%).
The following approaches could further improve the paper:
-
Extract the SQL requests formed by each ORM, present them in the paper, and compare them.
-
Change the database implementation to the fake one (mock) and evaluate the performance of each isolated ORM layer.
-
Some specific real-life scenarios could be analyzed (e.g., mass addition of items, logging, or mass change of items) and evaluated to see which mappings are more effective.
There is also some criticism of the presentation of the information in the paper.
-
The abstract should include the key findings of the paper - some facts supported by numbers that were most remarkable in the research. It should also clearly state the study's final goal (in one sentence).
-
The introduction could include more cited research. Several more research publications dedicated to database performance measurement and framework could be noted in the "Previous research" section.
-
Graphs show absolute values only, which makes the values harder to compare. It can be good to give the time per operation in nanoseconds for different numbers of executions.
-
There are no aggregated tables and graphs to compare the whole set of tested ORM-DB combinations.
-
There are too many figures and tables in sections 7-8 (Results), which should be moved into supplemental material. Only the most important figures and tables thoroughly described in the text should be left in the results section.
-
The last part of the conclusion section should also include a clear and distilled take-home message.
The language in the paper sounds non-scientific, with some blogger-type phrases that look inappropriate for scientific papers. E.g., "Having no clue about frameworks and just using them on other people's recommendations is not a great way to start an application" does not sound like a statement with a presented evidence base behind it. Another example: "Usually, people tend to choose the technologies they know best.". Or "while READ is really fast, under one second even for the biggest test," is somewhat a subjective judgment, as well as "READ is also pretty good."
The authors should consider rewriting parts written in somewhat colloquial and entertaining language in favor of a scientific style.